# Sub-diurnal asymmetric warming has amplified atmospheric dryness since the 1980s

Ziqian Zhong ®¹, Hans W. Chen ®¹ ✉, Aiguo Dai ®², Tianjun Zhou ®³, Bin He ®⁴ & Bo Su⁴,⁵,⁶

Rising atmospheric vapor pressure deficit (VPD)—a measure of atmospheric dryness, defined as the difference between saturated vapor pressure (SVP) and actual vapor pressure (AVP)—has been linked to increasing daily mean near-surface air temperatures since the 1980s. However, it remains unclear whether the faster increases in daily maximum temperature ($T_{max}$) relative to daily minimum temperature ($T_{min}$) have contributed to rising VPD. Here, we show that the faster rise in $T_{max}$ compared with $T_{min}$ over land has intensified VPD from 1980 to 2023. This sub-diurnal asymmetric warming has driven a larger SVP increase than would occur under uniform temperature rise, while AVP is more strongly influenced by $T_{min}$. Using reanalysis data, we estimate that asymmetric warming has contributed an additional ~18% to the increase in global land VPD. Sub-daily station observations corroborate this pattern, with asymmetric warming accounting for ~30% of VPD intensification across all stations. Our findings indicate that sub-diurnal asymmetric warming has substantially amplified global warming's effect on atmospheric dryness over the past four decades, with significant implications for terrestrial water availability and carbon cycling.

Global surface temperatures have been rising, with warming accelerating across nearly all continents in recent decades[1,2]. The past nine years, from 2015 to 2023, have been the warmest on record[3–5]. Rising air temperatures increase the near surface (~2 m) saturation pressure (SVP)—the atmosphere's capacity to hold water vapor—by roughly 7% per 1 °C warming according to the Clausius-Clapeyron relationship[6,7]. As the near surface actual vapor pressure (AVP) has generally increased at a lower rate than SVP, their difference, known as the atmospheric vapor pressure deficit (VPD), has increased in all climatic zones since 1980s[8–10]. The notable VPD increase in recent decades has significantly affected vegetation growth and productivity[11–14],

maize yield[15,16], land evapotranspiration[17,18], and the occurrence of wildfires[19,20] worldwide.

VPD changes are controlled by both atmospheric temperature and moisture variations. This is because SVP is almost exclusively determined by air temperature, while AVP depends on both temperature and moisture variations. Another common metric for the water vapor content in the atmosphere is relative humidity (RH), defined as the ratio of AVP to SVP. Unlike the increase in SVP due to rising air temperatures, the long-term trend of globally averaged near-surface RH over land has remained small until the early 2000s[21], and a significant decline has been reported after the year 2000[22,23]. The

¹Department of Space, Earth and Environment, Division of Geoscience and Remote Sensing, Chalmers University of Technology, SE-412 96 Gothenburg, Sweden. ²Department of Atmospheric and Environmental Sciences, University at Albany, State University of New York, Albany, NY 12222, USA. ³State Key Laboratory of Numerical Modeling for Atmospheric Sciences and Geophysical Fluid Dynamics, Institute of Atmospheric Physics, Chinese Academy of Sciences, Beijing 100029, China. ⁴State Key Laboratory of Earth Surface Processes and Resource Ecology, Faculty of Geographical Science, Beijing Normal University, Beijing 100875, China. ⁵Regional Climate Group, Department of Earth Sciences, University of Gothenburg, S-40530 Gothenburg, Sweden. ⁶Stockholm Resilience Centre, Stockholm University, 10691 Stockholm, Sweden. ✉e-mail: hans.chen@chalmers.se

factors contributing to the spatial patterns of RH trends on a global scale remain unclear[24]. Additionally, although surface air temperature influences both SVP and RH, and is a key climatic factor affecting VPD, the impact of sub-daily temperature variations on VPD is often overlooked, even though diurnal temperature variations have been found to be the main driver of RH diurnal variations[21].

Given the approximately exponential relationship between air temperature and SVP[25], a temperature increase during warmer daytime hours results in a larger SVP increase than an equivalent warming during cooler nights. Recent studies have found that daily maximum temperatures ($T_{max}$) have increased more rapidly than daily minimum temperatures ($T_{min}$) over land since the 1980s[26,27]. This sub-diurnal asymmetric warming is primarily driven by changes in solar radiation, which is closely linked to variations in cloud cover and aerosol emissions[26,28,29]. The resulting increase in the diurnal temperature range (DTR = $T_{max}$ − $T_{min}$) marks a shift from previously faster nighttime warming to faster daytime warming[30–33]. This shift raises the question: What is the relative importance of $T_{max}$ and $T_{min}$ in driving VPD variations?

To answer this question, we employed ridge regression[34] to assess the effects of interannual variations in $T_{max}$ and $T_{min}$ on VPD. Considering the potential complex interactions and non-linear relationships among the environmental factors examined, we further applied a Random Forest (RF) regression model[35] enhanced with Shapley Additive Explanations[36] (RF-SHAP) as a complementary approach. The RF approach delivers robust predictive performance by effectively capturing intricate, non-linear data patterns that traditional linear models may not adequately capture. In addition, the SHAP framework provides a clearer understanding of each environmental factor's relative contribution. Finally, using flux tower observations, we examined daily temperature–VPD relationships for both $T_{max}$ and $T_{min}$, then implemented RF regression to evaluate the role of sub-diurnal asymmetric warming in driving long-term VPD changes.

## Results
### Interannual analysis
In our study, we used both sub-daily in-situ observations from the HadISD dataset[37] and hourly data from the ERA5-Land reanalysis[38]. Previous research has shown that the ERA5 reanalysis effectively captures diurnal variations in climatic variables[39,40]. However, in some regions—such as East Asia—correlations between DTR values derived from the ERA5-Land reanalysis and co-located HadISD observations are notably weaker (see Supplementary Discussion 1, Supplementary Fig. 1). Therefore, most of our analyses were conducted using both the HadISD and ERA5-Land datasets in parallel, to provide complementary insights and mutual support.

We first identified the annual trends in DTR and VPD over the period 1980–2023. VPD can be calculated using the following equation[41]:

$$VPD = SVP \times (1 - RH) \tag{1}$$

The significant ($p < 0.05$) increase in annual-mean VPD (Supplementary Fig. 2a), at an average rate of 0.24 hPa per decade across 1398 stations, can thus be attributed to a substantial rise in SVP and a significant decrease in RH (Supplementary Fig. 2c). Previous regional studies have suggested that RH measurements may exhibit discontinuities due to recent hygrometer changes[42]. However, these discontinuities are localized and limited in number, and the associated uncertainties are essentially negligible when assessing large-scale characteristics of RH[43]. The positive VPD trends were most pronounced in the mid- to low-latitude regions based on the ERA5-Land reanalysis dataset (Supplementary Fig. 3f).

Due to the faster increase in $T_{max}$ than in $T_{min}$, the mean DTR across all stations has significantly increased at an average rate of

0.10 °C per decade. Spatially, based on the ERA5-Land dataset, we found that 58% of global land areas experienced an increase in DTR. Notably, the area with a significant increase in DTR (30%) was nearly twice as large as the area with a significant decrease (15%) during 1980–2023 (Fig. 1a). This finding is consistent with previous studies reporting increasing DTR in recent decades[26,27]. Regions showing DTR increases were mainly located in western North America, South America, Europe, central Africa, Central Asia, eastern China, and Australia. We found a strong correlation between the annual average DTR and VPD during 1980–2023 (correlation coefficient $r = 0.88$, $p < 0.05$) across all stations (Supplementary Fig. 2a). This correlation remained strong and significant ($r = 0.64$, $p < 0.05$) after removing their long-term linear trends. The ERA5-Land reanalysis dataset also showed similar increasing trends in annual mean DTR and VPD over land (Supplementary Fig. 2d), along with a significant correlation between them ($r = 0.76$, $p < 0.05$ for the raw time series and $r = 0.60$, $p < 0.05$ after detrending).

We then investigated the impact of interannual variations in DTR on VPD. Specifically, we employed ridge regression[34] and an RF regression model, using DTR, daily mean temperature ($T_{mean}$), and soil moisture (SM) as independent variables, and VPD as the dependent variable (Supplementary Fig. 4):

$$VPD \sim f(DTR, T_{mean}, SM) \tag{2}$$

where $f$ represents the functional relationship between VPD and the independent variables. In addition to this formulation, VPD was also modeled as a function of $T_{max}$, $T_{min}$, and SM:

$$VPD \sim f(T_{max}, T_{min}, SM) \tag{3}$$

SM was included as an independent variable in both equations because it is a key source of surface vapor and physically influences VPD through its role in evaporation. Moreover, SM can modulate DTR by enhancing evaporative cooling, which typically has a stronger effect on $T_{max}$ than $T_{min}$[44]. Therefore, it is essential to account for the influence of SM when assessing the DTR–VPD relationship in the regression models. Based on the ridge regression model defined in Eq. (2) and detrended, standardized interannual variables, we found that interannual fluctuations in VPD were positively associated with DTR variations across both station-based observations and ERA5-Land grid points during 1980–2023 (Fig. 1b). We further applied the ridge regression model defined in Eq. (3) to assess the differential impacts of sub-daily temperatures on VPD (Supplementary Fig. 5). Overall, $T_{max}$ exhibited a strong positive influence on interannual VPD variability (Fig. 1c), whereas $T_{min}$ showed a relatively weaker effect across both station-based observations and ERA5-Land grid points (Fig. 1e). We further estimated the relative importance of $T_{max}$ and $T_{min}$ in driving interannual VPD variability using the RF-SHAP framework (Supplementary Fig. 6). This approach mitigates issues of multicollinearity and captures nonlinear interactions. Relative importance was calculated as the normalized magnitude of the absolute SHAP values, which represent the importance of each predictor in the RF regression model. The relative importance of $T_{max}$ accounted for a median of 41% of VPD variability across all stations (Fig. 1d) and 40% across all ERA5-Land grid points (Fig. 1f), whereas the relative importance of $T_{min}$ was notably lower, with a median of 18% across all stations (Fig. 1d) and 22% in ERA5-Land (Fig. 1f). These results indicate an asymmetric effect of $T_{max}$ and $T_{min}$ on atmospheric dryness, with $T_{max}$ playing a more dominant role than $T_{min}$ in driving the interannual variability of VPD.

VPD can be expressed as either VPD = SVP − AVP or VPD = SVP × (1 − RH) (Eq. 1), indicating that VPD variations depend on changes in SVP and either AVP or RH. Using detrended interannual daily mean variables, we conducted further analyses based on three regression

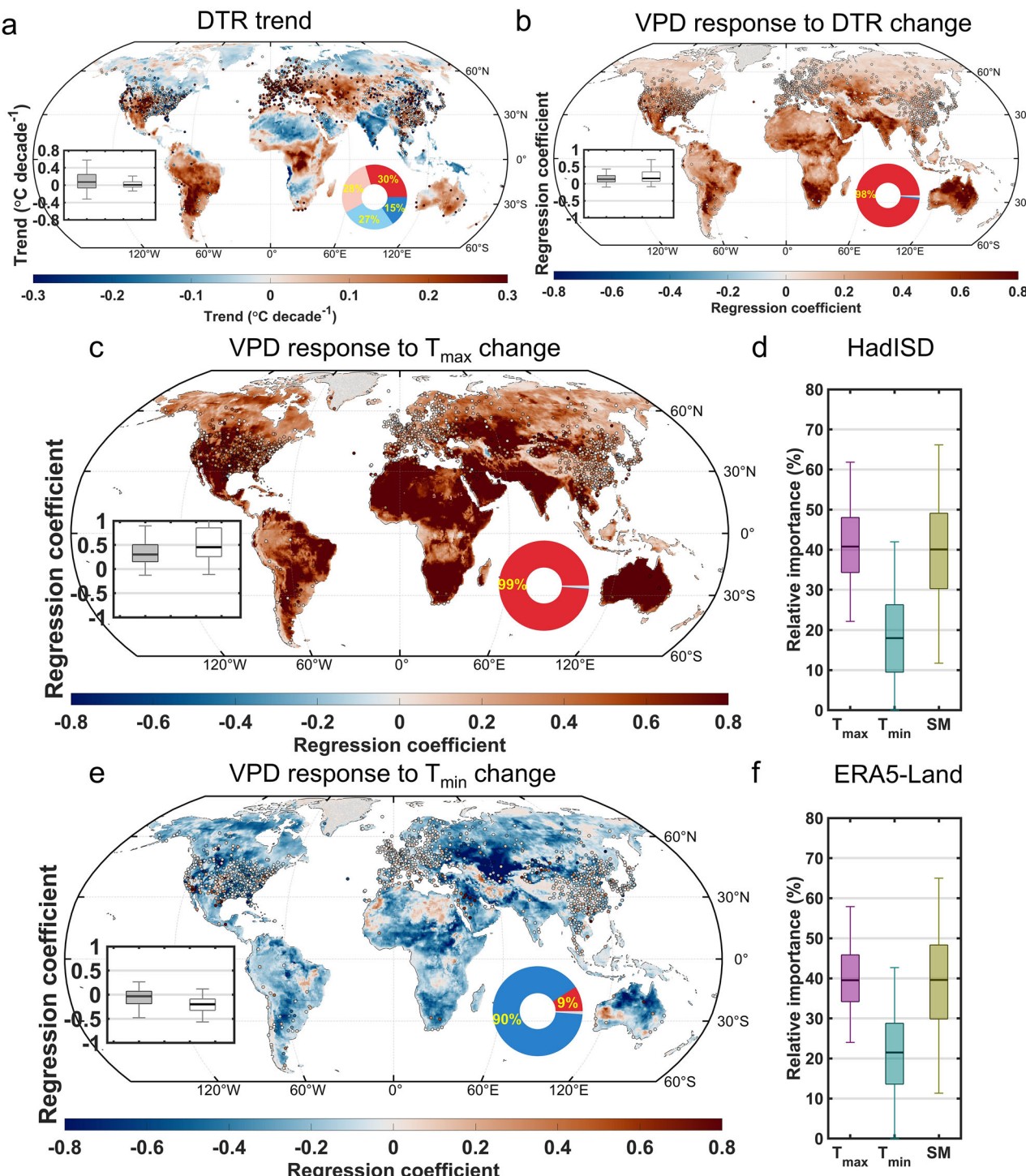

**Fig. 1 | Relative importance of daily maximum ($T_{max}$) and minimum temperatures ($T_{min}$) in interannual vapor pressure deficit (VPD) variations from 1980 to 2023. a** Spatial distribution of the trend in diurnal temperature range (DTR) over land areas. Inset shows boxplots of trends across observation stations (solid boxes) and ERA5-Land grid points (hollow boxes). Pie chart shows the percentage of land area with significantly ($p < 0.05$) positive (red), weak positive (light red), weak negative (light blue), and significantly negative (blue) trends, based on ERA5-Land data. **b** Spatial distribution of ridge regression (RR) coefficients of interannual VPD with respect to DTR, derived from the RR model defined in Eq. (2). **c, e** Spatial distribution of RR coefficients of interannual VPD with respect to $T_{max}$ (**c**) and $T_{min}$ (**e**), derived from the RR model defined in Eq. (3). Insets in **b, c,** and **e** show boxplots of RR coefficients across observation stations (solid boxes) and grid points (hollow

boxes). Pie charts show the percentage of land area with positive (red), negative (blue), and non-significant (light grey) RR coefficients based on ERA5-Land data. "Non-significant" refers to cases where none of the coefficients in the RR model are statistically significant. Stations (1.07% in b; 1.29% in **c, e**) and areas (1.10% in b; 1.11% in c, e) with non-significant RR coefficients are masked in light grey and excluded from RR model-based analysis. **d, f** Relative importance of interannual $T_{max}$, $T_{min}$, and soil moisture (SM) in driving interannual VPD variability based on the Random Forest regression model defined in Eq. (3), evaluated across stations (**d**) and ERA5-Land grid points (**f**). All variables in the regressions are detrended and standardized annual averages. In all boxplots, the height of each box represents the interquartile, with the thick black line indicating the median, and the edges denoting the first and third quartiles. Whiskers extend to the 2.5th and 97.5th percentiles.

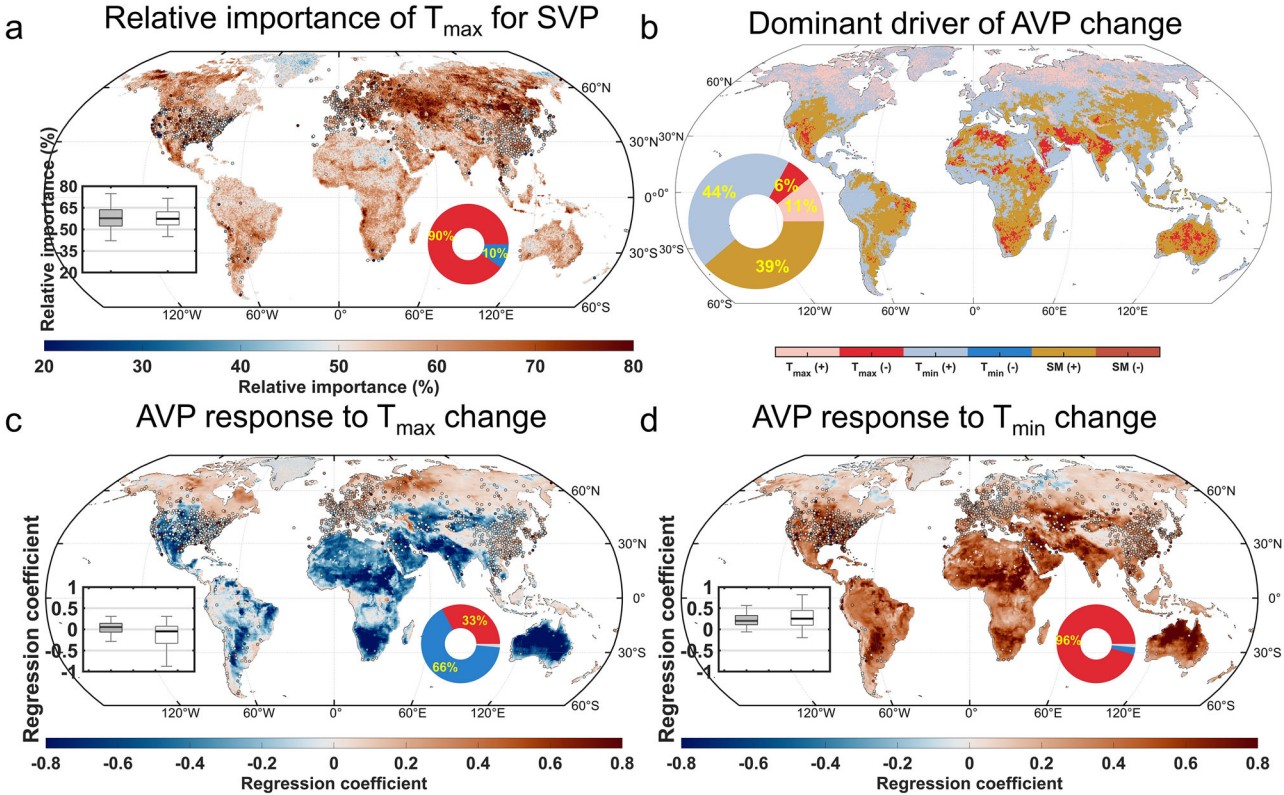

**Fig. 2 | Relative importance of daily maximum ($T_{max}$) and minimum temperatures ($T_{min}$) in interannual variations of saturation vapor pressure (SVP) and actual vapor pressure (AVP) from 1980 to 2023. a** Spatial distribution of relative importance of $T_{max}$ in interannual SVP, identified using the Random Forest (RF) regression model defined in Eq. (4) with the Shapley Additive Explanations (SHAP) framework. Inset shows boxplots of relative importance (%) across stations (solid boxes) and ERA5-Land grid points (hollow boxes). Pie chart shows the percentage of land area where $T_{max}$ importance exceeds 50% (red) or is below 50% (blue), based on ERA5-Land data. **b** Spatial distribution of the dominant driver of interannual AVP changes, identified using the RF regression model defined in Eq. (5) with the SHAP framework, based on ERA5-Land data. Pie chart shows the percentage of land area where AVP is dominantly driven by $T_{max}$, $T_{min}$, or soil moisture (SM), with either a positive (+) or negative (−) influence. **c** and **d** spatial distribution of ridge regression

(RR) coefficients of interannual AVP with respect to $T_{max}$ (**c**) and $T_{min}$ (**d**), derived from the RR model defined in Eq. (5). Insets show boxplots of RR coefficients across observation stations (solid boxes) and grid points (hollow boxes). Pie charts show the percentage of land area with positive (red), negative (blue), and non-significant (light grey) RR coefficients based on ERA5-Land data. "Non-significant" refers to cases where none of the coefficients in the RR model are statistically significant. Stations (4.36%) and areas (1.45%) with non-significant RR coefficients are masked in light grey and excluded from the analysis. All variables in the regressions are detrended and standardized annual averages. In all boxplots, the height of each box represents the interquartile, with the thick black line indicating the median, and the edges denoting the first and third quartiles. Whiskers extend to the 2.5th and 97.5th percentiles.

models:

$$SVP \sim f(T_{max}, T_{min}) \tag{4}$$

$$AVP \sim f(T_{max}, T_{min}, SM) \tag{5}$$

$$RH \sim f(T_{max}, T_{min}, SM) \tag{6}$$

Based on the RF regression model defined in Eq. (4) and the RF-SHAP framework (Supplementary Fig. 7), we found that the inter-annual variations in SVP were primarily determined by $T_{max}$, with $T_{max}$ exerting a greater impact on SVP than $T_{min}$ across nearly 90% of land areas in the ERA5-Land dataset (Fig. 2a). The stronger influence of $T_{max}$ on SVP is due to the near-exponential relationship between temperature and SVP, meaning the rate of SVP increase is greater at higher temperatures. For instance, according to the Clausius-Clapeyron relationship, while a 1 °C increase at global average land $T_{min}$ (8.6 °C) leads to a 0.78 hPa rise in SVP, the same temperature increase at global annual average land $T_{max}$ (18.2 °C) results in a 1.35 hPa rise—roughly 72% higher (Supplementary Fig. 8a). This differential effect is even more pronounced in low-to-mid-latitudes, where a 1 °C increase at

typical land $T_{max}$ (25.5 °C) produces a 76% higher SVP compared to $T_{min}$ (15.8 °C). This illustrates why $T_{max}$ exerts a stronger influence on SVP than $T_{min}$, and indicates that in warmer conditions, the difference in the impact of $T_{max}$ and $T_{min}$ on SVP will become even more pronounced.

Based on the ridge regression model defined in Eq. (5), we found that $T_{min}$ has significantly affected AVP across 93% of the land area based on the ERA5-Land dataset (Supplementary Fig. 9d). Overall, $T_{min}$ has exerted a positive influence on AVP, with 96% of land area exhibiting positive regression coefficients based on the ERA5-Land dataset (Fig. 2d). Using the RF-SHAP framework with the same independent and dependent variables (Supplementary Fig. 10), we found that $T_{min}$ was the dominant driver of AVP over 44% of land areas, primarily located in the mid-latitudes, where it has exerted a positive influence (Fig. 2b). Here, the "dominant driver" refers to the variable that contributes most to the RF regression model output, as indicated by the highest mean absolute SHAP value among all predictors. The direction of influence—positive or negative—is determined by the Theil–Sen slope between the variable's values and their corresponding SHAP values. In contrast, $T_{max}$ was the dominant driver with a positive influence over only 11% of land areas, and with a negative influence over 6%. This widespread positive influence of $T_{min}$ on AVP may also

have contributed to its broad positive effect on RH. According to ridge regression results from Eq. (6), 95% of grid cells had positive regression coefficients for the effect of $T_{min}$ on RH based on the ERA5-Land dataset (Supplementary Fig. 11d). The positive effect of $T_{min}$ on AVP (Fig. 2d) can be supported by the general alignment of dew point temperature with $T_{min}$[45], along with the near-exponential relationship between AVP and dew point temperature according to the Clausius-Clapeyron relationship. Air temperature governs the maximum amount of water vapor that the atmosphere can hold. Typically, over land—especially under more humid conditions—RH often approaches 100% around the time of $T_{min}$, while AVP (or specific humidity) is relatively stable through the 24-hr diurnal cycle[21]. This suggests that $T_{min}$ largely controls the water vapor content or AVP in the air when it is close to saturation. Although slight diurnal fluctuations in water vapor content may occur, $T_{min}$ plays a more substantial role than $T_{max}$ in controlling the daily-mean water vapor content (Supplementary Fig. 8b).

We found that the positive response of AVP to $T_{max}$ was relatively heterogeneous and regionally variable compared to its response to $T_{min}$ (Fig. 2b, c). During the daytime, increased temperature typically elevates VPD, which enhances land evapotranspiration over wet surfaces[46], thereby increasing atmospheric water vapor. However, over drylands, the relationship between temperature/VPD and evapotranspiration becomes more complex, particularly when considering plant physiological processes[47]. Using half-hourly observational data from FLUXNET tower sites[48] (Supplementary Fig. 12), we found that the diurnal cycle of VPD closely mirrors that of temperature ($R^2 = 0.985 \pm 0.014$; mean ± one standard deviation across all sites). This similarity arises because SVP, which is strongly temperature-dependent ($R^2 = 0.997 \pm 0.002$), exhibits greater diurnal variation than AVP, with daily standard deviations of 2.76 hPa versus 0.17 hPa, respectively (Supplementary Fig. 13). Consequently, higher $T_{max}$ produces greater daily maximum VPD. During high VPD conditions, plant stomata tend to partially close in response to increased atmospheric dryness[49,50]. This "feed-forward" response[51] reduces transpiration rates under high VPD conditions[25], thereby limiting increases in AVP. The inhibitory effect is particularly pronounced in water-limited areas of low- to mid-latitudes (Fig. 2c), where the climate is relatively hot and dry.

## Daily analysis

The above findings suggest that changes in $T_{max}$ and $T_{min}$ have different impacts on daily-mean VPD. When $T_{max}$ increases, it will lead to a larger increase in SVP than AVP and result in a noticeable increase in VPD. In contrast, for an equivalent increase in $T_{min}$, the resulting increase in SVP is smaller, while RH will increase in most regions, thus partially offsetting the increase in VPD associated with rising $T_{max}$. Building on observational data from FLUXNET tower sites, we further investigated the impact of diurnal temperatures on VPD after accounting for the influence of SM on the relationship between DTR and VPD. We conducted this analysis because increased SM can enhance evaporative cooling, which reduces $T_{max}$ and consequently lowers DTR[44]. Simultaneously, the increased SM raises AVP and RH, further decreasing VPD[52,53]. These processes could lead to a positive correlation between DTR and VPD.

To mitigate the influence of SM, we segmented the daily data into different bins based on the percentiles of the surface soil water content (SWC) within each flux tower site. Before the analysis, we employed Fourier transform-based filtering[54] (Supplementary Fig. 14) to remove seasonal variations from the daily variables. In all bins, the correlation between SWC and either VPD or DTR was generally weak, indicating decoupling[50,55] between SWC and VPD or DTR within each bin (Supplementary Fig. 15a). We then compared the partial correlations between SVP, AVP, and RH with $T_{max}$ or $T_{min}$ within each bin (Fig. 3). In the partial correlation analysis, we accounted for the influence of $T_{min}$

when estimating the correlations between $T_{max}$ and SVP, AVP, or RH. Similarly, we controlled for $T_{max}$ when calculating correlations with $T_{min}$. Across all bins, $T_{max}$ showed a stronger correlation with SVP than $T_{min}$, while $T_{min}$ exhibited a stronger positive correlation with AVP than $T_{max}$. Since RH is the ratio between AVP and SVP, $T_{max}$ was generally negatively correlated with RH, while $T_{min}$ showed a positive correlation with RH. These findings, when considered alongside Eq. (1), indicate a strong positive correlation between VPD and $T_{max}$, and a weak positive or even negative correlation between VPD and $T_{min}$. This is consistent with the results of the partial correlation analysis between VPD and $T_{max}$ or $T_{min}$ (Supplementary Fig. 15b). These results further demonstrate the asymmetric effect of $T_{max}$ and $T_{min}$ on VPD, even after accounting for soil moisture effects.

## Trend analysis

The preceding analysis reveals asymmetric effects of $T_{max}$ and $T_{min}$ on VPD at both interannual and daily scales. To further quantify the long-term impact of sub-diurnal asymmetric warming on VPD changes, we employed monthly, non-detrended variables—including DTR, $T_{mean}$, and SM—in an RF regression model as defined in Eq. (2) to predict monthly VPD values from 1980 to 2023. This RF approach was specifically chosen to capture the complex nonlinear relationships among these variables. We first trained the model to optimize its parameters, and then applied the trained model on the input data to obtain fitted VPD values ($VPD_{fitted}$). The median out-of-bag $R^2$ for the RF models reached 0.91 for the HadISD dataset and 0.96 for the ERA5-Land dataset (Supplementary Fig. 16a), indicating that the models effectively capture most of the variance in VPD over land. Subsequently, three sensitivity experiments (see Methods) were conducted, one for each independent variable, keeping the tested variable constant at its mean value for each month during the control period (the initial three years, 1980–1982), while the other two variables varied according to the input. The difference between $VPD_{fitted}$ and the estimated VPD from each sensitivity experiment was considered the contribution of DTR, $T_{mean}$, and SM change to the VPD change, denoted as $VPD_{DTR}$, $VPD_{Tmean}$, and $VPD_{SM}$, respectively.

On average, across all stations from 1980 to 2023, $VPD_{DTR}$ increased at a rate of 0.06 hPa per decade ($p < 0.05$, Fig. 4a). An upward trend in $VPD_{DTR}$ was observed at 80% of the stations, with 45% showing a statistically significant increase ($p < 0.05$, Fig. 4d). We then focused on the contribution rate of DTR change to the VPD increase, defined as the ratio of the slopes of $VPD_{DTR}$ and $VPD_{fitted}$. On average, this contribution rate reached approximately 30% across all stations (Fig. 4a). Using the ERA5-Land dataset, DTR changes contributed an average of 18% to the VPD increase across land areas (Supplementary Fig. 17a). For the spatial analysis, we concentrated on stations where $VPD_{fitted}$ exhibited a significant increase ($p < 0.05$), representing about 66% of all stations. Here, the median contribution rate of DTR change to VPD increase reached 35% (Supplementary Fig. 18). According to ERA5-Land, the median contribution rate reached 22% in regions where VPD showed a significant increase. These results indicate that the increase in DTR has played a notable role in promoting the rise in VPD since the 1980s.

To further investigate the potentially distinct long-term impacts of $T_{max}$ and $T_{min}$ increases on VPD, we quantified the contributions of changes in monthly variables to SVP (Supplementary Fig. 19) and RH (Supplementary Fig. 20) changes using RF models as defined in Eqs. (4) and (6), respectively. On average, the contribution of $T_{max}$ to SVP ($SVP_{Tmax}$) was greater than that of $T_{min}$ ($SVP_{Tmin}$). The growth rate of $SVP_{Tmax}$ exceeded that of $SVP_{Tmin}$ in most mid-latitude regions, including southwestern North America, central and eastern South America, southern Europe, central Africa, Central Asia, and Australia (Fig. 4e). The contribution of $T_{max}$ to RH showed a significant decreasing trend (−0.33% per decade, $p < 0.05$, based on observations, Fig. 4c; and −0.34% per decade, $p < 0.05$, over land in the ERA5-Land

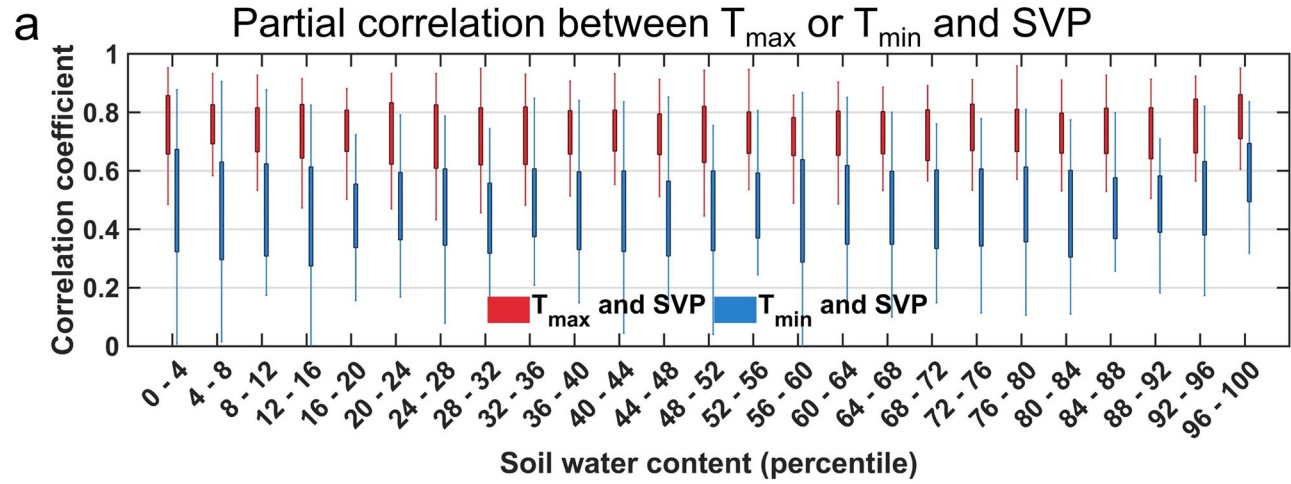

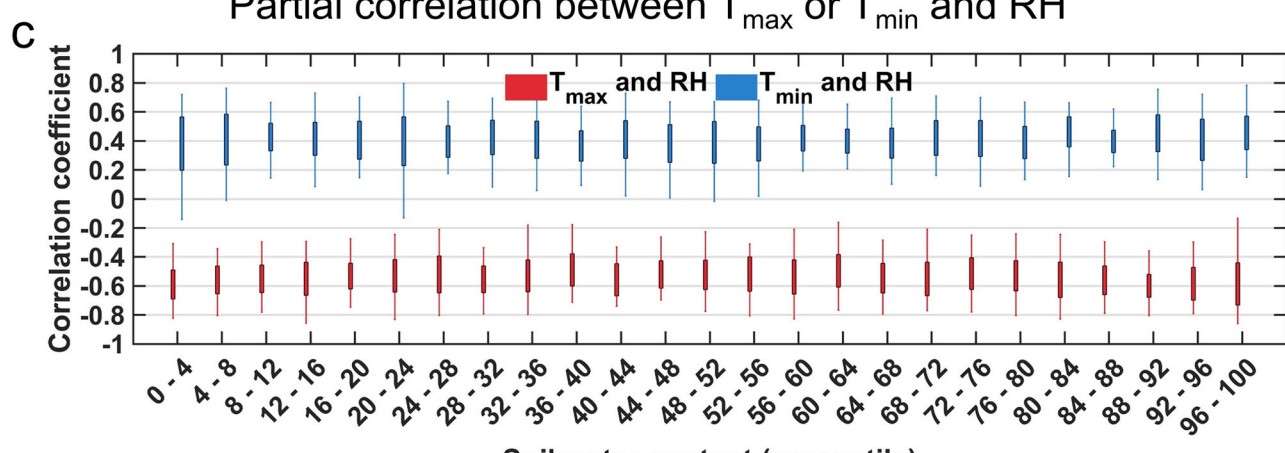

**Fig. 3 | Asymmetric effects of daily maximum ($T_{max}$) and minimum temperatures ($T_{min}$) on saturated vapor pressure (SVP), actual vapor pressure (AVP), and relative humidity (RH).** Assessment of the partial correlation between $T_{max}$ or $T_{min}$ and SVP (**a**), AVP (**b**), and RH (**c**) while controlling for the other variable across different soil water content percentiles at all FLUXNET sites. The height of each box represents the interquartile range of correlation coefficients across different stations, with the edges denoting the first and third quartiles. Whiskers extend to the 2.5th and 97.5th percentiles of the correlation coefficient.

reanalysis, Supplementary Fig. 17c), which was widespread across most of the land areas except for northern North America and India. In contrast, the contribution of $T_{min}$ to RH exhibited a slight but significant increasing trend (0.13% per decade, $p < 0.05$, across stations; and 0.13% per decade, $p < 0.05$, over land in the ERA5-Land dataset), which was prevalent across most land areas except for the western

United States and Central Asia (Fig. 4g). This analysis reinforces that over the past few decades, changes in $T_{max}$ and $T_{min}$ have had different effects on both SVP and RH. Generally, increases in $T_{max}$ have had a bigger impact on increasing SVP than $T_{min}$. Additionally, while increases in $T_{min}$ generally appear to have a positive effect on RH, increases in $T_{max}$ could contribute to decreased near-surface RH over

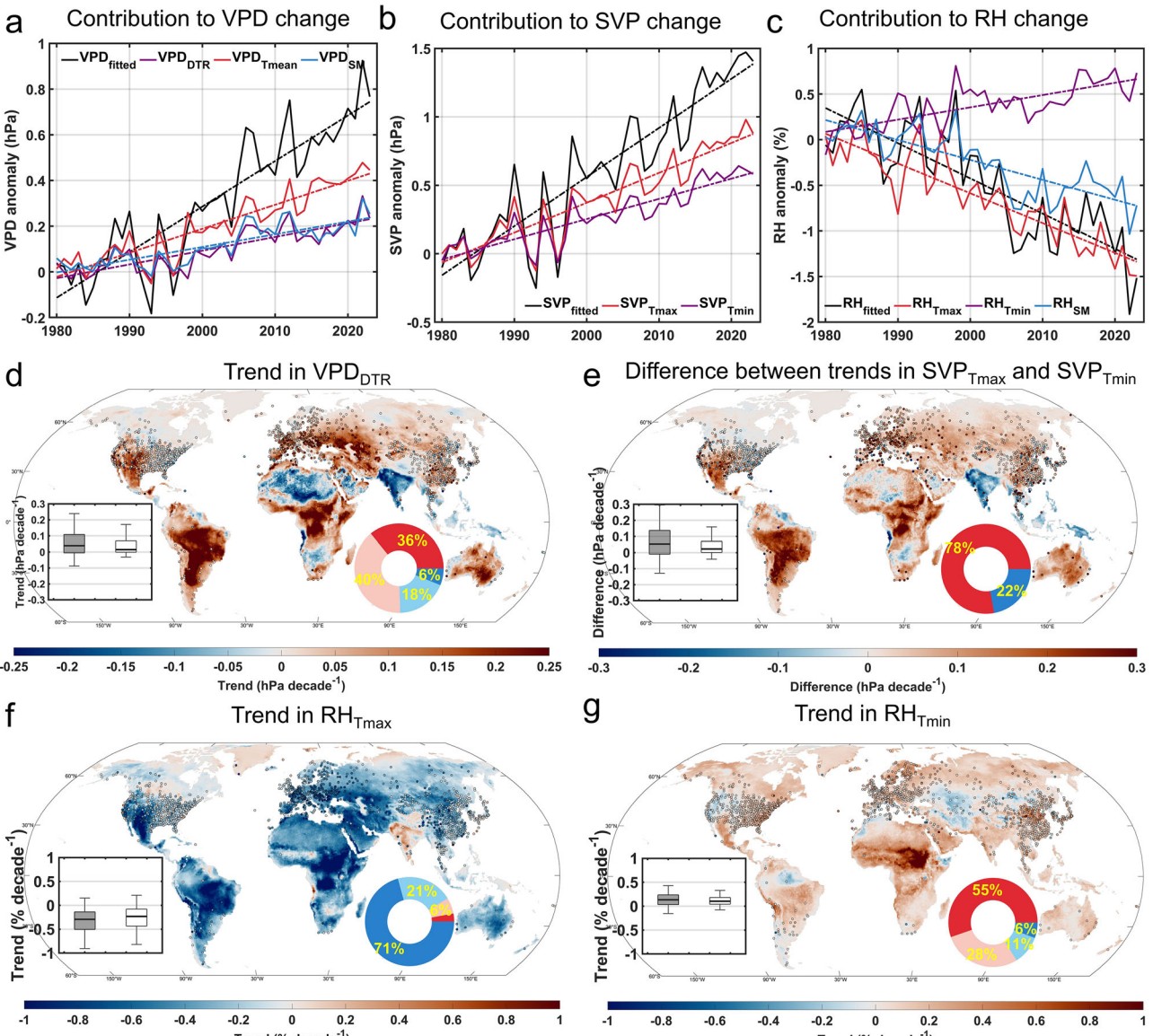

**Fig. 4 | Contribution of sub-diurnal asymmetric warming to trends in saturated vapor pressure (SVP), relative humidity (RH), and vapor pressure deficit (VPD) from 1980 to 2023. a–c** Variations and changes in the annual average model-fitted (subscript fitted) VPD (**a**), SVP (**b**), and RH (**c**), and the contributions of diurnal temperature range (subscript DTR), daily mean temperature (subscript $T_{mean}$), soil moisture (subscript SM), daily maximum temperature (subscript $T_{max}$) and daily minimum temperature (subscript $T_{min}$) to the variations and changes over land. The model-fitted values are anomalies calculated by subtracting the mean values for the control period (1980–1982). The dashed lines show the linear trends obtained from linear regressions. **d**, **f** and **g** Spatial distribution of trends in $VPD_{DTR}$ (**d**), $RH_{Tmax}$ (**f**), and $RH_{Tmin}$ (**g**). Insets show boxplots of trends across observation

stations (solid boxes) and ERA5-Land grid points (hollow boxes). Pie charts show the percentage of land area with significantly ($p < 0.05$) positive (red), weak positive (light red), weak negative (light blue), and significantly negative (blue) trends, based on ERA5-Land data. **e**, Spatial distribution of differences between trends in $SVP_{Tmax}$ and $SVP_{Tmin}$. Insets show boxplots of differences across observation stations (solid boxes) and ERA5-Land grid points (hollow boxes). Pie chart show the percentage of land area with positive (red) and negative (blue) differences based on ERA5-Land data. In all boxplots, the height of each box represents the interquartile range of trends or differences across different stations or grid points, with the thick black line indicating the median, and the edges denoting the first and third quartiles. Whiskers extend to the 2.5th and 97.5th percentiles.

many land regions (Supplementary Fig. 11c). Due to these dual effects, the asymmetric warming characterized by daytime warming and increasing DTR over the past forty years has exacerbated atmospheric dryness over most land areas.

These results raise important questions that warrant further exploration: did VPD decline prior to 1980, and if so, was it related to faster nighttime warming observed during that period? To investigate this question, we extended the study period back to 1950 using the ERA5-Land dataset. We found that, over land, VPD significantly declined from the 1950s to the mid-1970s at a rate of −0.25 hPa per decade ($p < 0.05$), reaching a minimum around 1976 (Supplementary

Fig. 21). This decline was primarily associated with a significant decrease in $T_{max}$ (−0.13 °C per decade, $p < 0.05$), while $T_{min}$ showed little change (−0.03 °C per decade, $p > 0.05$) during 1950–1976. In contrast, during the period from 1977 to 2023, VPD increased significantly at a rate of 0.31 hPa per decade ($p < 0.05$), in parallel with a faster warming of $T_{max}$ (0.30°C per decade, $p < 0.05$) compared to $T_{min}$ (0.27 °C per decade, $p < 0.05$). We further performed partial correlation analyses using annual land-average variables during each period. The partial correlation between $T_{max}$ and VPD was 0.69 ($p < 0.05$) for 1950–1976 and 0.62 ($p < 0.05$) for 1977–2023, after controlling for $T_{min}$ and SM. These correlations were stronger than those between $T_{min}$ and

VPD during the same periods (−0.60 for 1950–1976 and −0.46 for 1977–2023, $p < 0.05$), controlling for $T_{max}$ and SM. Together, these findings reinforce the robustness of our conclusions, highlight the asymmetric effects of $T_{max}$ and $T_{min}$ on atmospheric dryness, and underscore the dominant role of $T_{max}$ in driving long-term changes in VPD.

## Implications for drought and wildfire risk

Increased atmospheric dryness directly contributes to higher atmospheric evaporative demand, which has been identified in a recent study as playing an increasingly important role in the occurrence of severe droughts[56]. To explore the relationship between atmospheric dryness and drought, we analyzed the correlation between annual VPD and the self-calibrated Palmer Drought Severity Index (scPDSI)[57] during 1980–2023. As a standardized drought index, lower values of scPDSI indicate more severe drought conditions. We found that VPD was significantly ($p < 0.05$) and negatively correlated with scPDSI across 47.7% of the global land area (Supplementary Fig. 22), suggesting a strong linkage between atmospheric dryness and drought conditions in these regions. These areas were mainly located in southwestern North America, eastern and southern South America, southern and eastern Europe, Central Asia, inland East Asia, and eastern Australia. Among the regions exhibiting significant negative VPD–scPDSI correlations, 68.6% experienced an increase in DTR and 69.3% experienced a decline in scPDSI. In contrast, among regions with either insignificant or positive VPD–scPDSI correlations, only 53.2% experienced increasing DTR. Furthermore, 62.9% of global land areas showed a significant negative correlation between DTR and scPDSI, with a spatial distribution pattern similar to that of regions with significant negative VPD–scPDSI correlations. These results indicate an important role of daytime warming in driving regional atmospheric drying and drought intensification[58]. Recent increases in drought severity or frequency reported in regions such as the southwestern United States[59,60], Europe[61,62], inner East Asia[63,64], and South America[65,66] may be closely linked to accelerated $T_{max}$ warming over recent decades (Fig. 1a).

Another major consequence of amplified atmospheric dryness is the increased frequency and severity of wildfires. Based on the fire weather index (FWI) from the European Centre for Medium-Range Weather Forecasts (ECMWF)[67], we found that a significant positive correlation exists between FWI and VPD (93.8% of land area) as well as between FWI and DTR (85.7% of land area) during 1980–2023 (Supplementary Fig. 23). These findings suggest a strong connection between faster daytime warming and heightened potential fire danger and intensity, as burned area is positively correlated with fire weather across much of the globe, including North and South America, Europe, and large parts of Asia[68]. Recent wildfire events in the southwestern United States[69,70], Mediterranean Europe[71], and South America[72] are likely linked to increases in both DTR and VPD since the 1980s (Fig. 1a and Supplementary Fig. 3f).

## Discussion

Our findings provide compelling evidence that stronger daytime warming over the past four decades has significantly contributed to increased atmospheric dryness. Given that the effects of SM on both $T_{max}$ and VPD are primarily mediated through evapotranspiration (ET), we conducted an additional ridge regression analysis by replacing SM with ET in Eqs. (3) and (5). Based on the ridge regression model defined as VPD ~ $f$ ($T_{max}$, $T_{min}$, ET), we found that $T_{max}$ has exerted a stronger positive influence on VPD than $T_{min}$ on the interannual scale. The spatial distribution of ridge regression coefficients (Supplementary Figs. 24a, 24b) is consistent with results from the original model defined in Eq. (3) (Fig. 1c, e), with spatial $r = 0.86$ and $r = 0.68$, respectively. Similarly, in the model defined as AVP ~ $f$ ($T_{max}$, $T_{min}$, ET), $T_{min}$ continued to show a more widespread and stronger positive effect on AVP compared to $T_{max}$ (Supplementary Fig. 25). The spatial distribution of regression

coefficients (Supplementary Figs. 25a, 25b) closely matches the corresponding results from the model based on Eq. (5) (Figs. 2c and 3d), with spatial $r = 0.84$ and $r = 0.86$, respectively. Previous studies suggest that ET is inherently difficult to measure accurately[73,74], particularly at large spatial scales[75], because it is influenced by a complex combination of environmental and biophysical factors[76]. Therefore, we used SM as a more reliable proxy in the main analysis.

Additionally, the negative response of AVP to $T_{max}$ in water-limited areas of low- to mid-latitudes (Fig. 2c) can be disrupted by the sunlight-blocking effect of clouds. Days with higher AVP typically exhibit increased cloud cover[77], which reflects incoming solar radiation and lowers surface incident radiation, consequently decreasing $T_{max}$. To mitigate this interference between AVP and $T_{max}$ due to radiation effects, we conducted an additional ridge regression analysis, derived from model defined as AVP ~ $f$ ($T_{max}$, $T_{min}$, SM, RS), where RS is surface incoming solar radiation, based on detrended and standardized variables from ERA5-Land from 1980 to 2023. The analysis reveals that the negative response of AVP to $T_{max}$ persisted across low- and mid-latitude regions (Supplementary Fig. 26a). The spatial distribution of the ridge regression coefficients closely aligns with the results from the analysis without including RS (Fig. 2c, spatial $r = 0.99$), indicating that the negative response of AVP to $T_{max}$ was not primarily caused by the sunlight-blocking effect of clouds. These analyses further confirm the robustness of our findings.

It is worth noting that SM increased significantly ($p < 0.05$) during 1950–1976 but showed a significant ($p < 0.05$) decline during 1977–2023 (Supplementary Fig. 27). This corresponds to the significant decreases in VPD and $T_{max}$ during the earlier period, and the significant increases in both variables in the later period. These contrasting trends raise another important question: since the 1980s, which effect has been dominant—the increase in $T_{max}$ leading to enhanced atmospheric dryness and then decreased SM, or the decline in SM reducing evaporative cooling and thereby contributing to higher $T_{max}$ and VPD? The recent increase in $T_{max}$ can be conceptually decomposed into two components. The first reflects the general warming trend shared with mean temperature and $T_{min}$, which is widely attributed to increased greenhouse gas concentrations. The second represents an additional increase in $T_{max}$ relative to $T_{min}$, primarily induced by radiation and mainly driven by widespread decreases in cloud cover and regional reductions in aerosol concentrations[26,28,29]. SM plays a negligible role in the first component and only a limited role in the second[26]. Therefore, SM decline is unlikely to be the main driver of the recent increases in $T_{max}$. Regarding the reason of SM decline, given that land precipitation has shown little long-term change[78]—suggesting relatively stable water input—the reduction in SM is more likely due to increased water loss from the soil, i.e., enhanced evaporation. Since increased VPD is a key driver of higher evaporative demand, this supports the interpretation that the dominant pathway is from increased $T_{max}$ leading to increased VPD and subsequently decreased SM, rather than the reverse.

The recent increase in VPD is clearly linked to global warming, and our findings reveal that this trend has been amplified by stronger daytime warming. Earth system models (ESMs) project a continued increase in VPD under future global warming scenarios[8,9]. However, the projected magnitude of this increase may be underestimated if the continued rise in surface solar radiation (i.e., "global brightening"[28,79,80]) and DTR, along with the amplification effect of sub-diurnal asymmetric warming identified in this study are not adequately accounted for. Moreover, the observed trend of increasing DTR since the 1980s has not been adequately captured in Coupled Model Intercomparison Project Phase 6 (CMIP6) models[27]. These findings underscore the urgent need to improve the simulation of future DTR trends and their influence on VPD, given that changes in atmospheric dryness could profoundly impact water cycling through land evapotranspiration[11], carbon cycling[81], and increase the frequency and

intensity of extreme events such as drought and wildfires[19,20], underscoring the critical need for heightened attention.

# Methods

## Data

HadISD[37] is an in-situ sub-daily (reporting from 6-hourly to hourly) dataset based on the NOAA ISD dataset[82]. Multiple quality checks were applied to the dataset, including the removal of duplicates, detection of distribution gaps, and identification of climatological outliers[37,83]. For station selection, we applied strict criteria: (1) valid days required minimum five paired observations of temperature and dew point temperature; (2) valid months allowed maximum 10 missing days and no more than 4 consecutive missing days[27]; (3) stations from 1980–2023 were included only if missing months were below 2%. After the selection, 1398 stations were retained for analysis. Considering the lack of SM measurements in HadISD, we used SM from the ERA5-Land[38] reanalysis at the station sites.

The observations from flux towers of hourly temperature, VPD, and soil water content were obtained from FLUXNET2015[48]. For SWC, we selected the shallowest measurement at a depth of 30 cm[84,85], representing the topsoil layer that directly interacts with the atmosphere. Only temperature, VPD, and SWC data with quality flags marked as "measured" or "good quality gapfill" were used. Time series encompassing complete observations over at least a three-year period were selected for analysis. After selection, 56 stations were retained for analysis, including 32 forest sites and 12 grassland sites. The forest sites included sites within deciduous broadleaf forests (DBF), evergreen needleleaf forests (ENF), evergreen broadleaf forests (EBF), and mixed forests (MF).

The hourly gridded temperature, dew point temperature, SM content of the soil layers and total evaporation were obtained from the ERA5-Land reanalysis dataset spanning 1950–2023 with a horizontal resolution of about 9 km. Here, SM content between 0 and 28 cm was calculated by summing up the moisture content for each layer weighted by the thickness of the layer[12]. Monthly scPDSI data[86] were obtained from the Climatic Research Unit (CRU), covering global land areas from 1980 to 2023 at a spatial resolution of 0.5° × 0.5°. The scPDSI was calculated based on CRU TS 4.08 precipitation and temperature data, combined with fixed parameters related to local soil and surface characteristics. The index ranges from −4 (extremely dry) to +4 (extremely wet), representing water supply and demand as determined by a complex water-budget model that incorporates soil properties, historical precipitation, and potential evapotranspiration. FWI data were obtained from the Copernicus Emergency Management Service (CEMS)[67], derived from ECMWF ERA5 reanalysis-based historical simulations, covering 1980 to 2023 at a spatial resolution of about 0.25° × 0.25°. The FWI combines the Initial Spread Index and the Build-Up Index to provide a numerical rating of potential frontal fire intensity and is widely used to inform the public about fire danger conditions.

All gridded datasets were aggregated to a common 0.5° × 0.5° grid before analysis.

## Calculation of vapor pressure deficit

VPD (hPa) was calculated using the following formula[6,7]:

$$VPD = SVP - AVP \tag{7}$$

$$SVP = 6.1078 \times e^{\frac{a \times T_a}{T_a + b}} \tag{8}$$

$$AVP = 6.1078 \times e^{\frac{a \times T_{dew}}{T_{dew} + b}} \tag{9}$$

Here, SVP is the saturated vapor pressure calculated based on the air temperature ($T_a$) in degrees Celsius. For $T_a$ at or above 0 °C, $a$ is

17.269 and $b$ is 237.3. For $T_a$ below 0 °C, $a$ is 21.875 and $b$ is 265.5[6]. AVP is the actual vapor pressure in hPa, which is defined as the vapor pressure of moist air at the ambient temperature or the saturation vapor pressure at the dew point[7,41]. $T_{dew}$ is the dew point temperature in degrees Celsius.

## Ridge regression and attribution

Ridge regression[34] minimizes the effects of high multicollinearity (i.e., correlation) among the independent variables, particularly by alleviating interference caused by strong correlations between temperature/DTR and SM[26]. The ridge regression was done separately using HadISD data at each station and ERA5-Land data at each grid box, with ERA5-Land SM data used in both analyses since HadISD does not include SM data. Prior to conducting the ridge regression analysis, long-term linear trends were removed from all variables. The detrended time series were then converted into z-scores by dividing the anomalies (from the linear trend) by their standard deviations for the period from 1980 to 2023. The ridge regression objective function ($\hat{\beta}$) can be expressed as follows[34]:

$$\beta^{\wedge} = \sum_{i=1}^{n} \left( y_i - \beta_0 - \sum \beta_j x_{ij} \right)^2 + \lambda \sum \beta_j^2 \tag{10}$$

where $n$ is the number of total data points, $y_i$ is the value of the dependent variable at time step $i$, $\beta_o$ is the intercept term, and $\beta_j$ signifies the regression coefficient for the independent variable $x_j$ whose value at $i$ time step is $x_{ij}$. The optimal regularization parameter ($\lambda$) was determined using leave-one-out cross-validation (LOOCV). LOOCV is a special case of $k$-fold cross-validation where $k$ (the number of subsamples) equals $n$ (the number of observations). LOOCV is particularly appealing for small datasets, as it maximizes the size of the training set in each iteration. Here, for each value of $\lambda$, we performed LOOCV by iteratively excluding one observation, fitting the ridge model to the remaining data, and computing the squared prediction error for the excluded observation. This procedure was repeated for all observations, ensuring that each data point served once as the validation set. The LOOCV error for that $\lambda$ was then calculated as the average of these squared errors. This procedure was repeated over a predefined set of $\lambda$ values, and the $\lambda$ with the lowest LOOCV error was selected as optimal. Formally, the LOOCV error for each $\lambda$ was computed as:

$$LOOCV\ error(\lambda) = \frac{1}{n} \sum_{i=1}^{n} \left( y_i - y_{(i)}^{\lambda} \right)^2 \tag{11}$$

where $y_{(i)}^{\lambda}$ is the prediction for the $i$-th observation using the model trained without the $i$-th sample.

To quantify the statistical significance of the ridge regression coefficients, we applied a nonparametric bootstrap approach with 1000 iterations. In each iteration, the original dataset was resampled with replacement, and the ridge regression model was refitted using the LOOCV-optimized $\lambda$. This resulted in a distribution of regression coefficients for each predictor, from which the 95% confidence intervals (CIs) were derived using the 2.5th and 97.5th percentiles. A coefficient was deemed statistically significant if its CI did not include zero. This approach allows for robust inference without assuming normally distributed residuals and accounts for sampling variability.

The Durbin–Watson statistic[87] was used to detect the presence of autocorrelation at lag 1 in the residuals (prediction errors) from the regression analysis. If $e_t$ denotes the residual at time $t$, the Durbin–Watson test statistic ($d$) is defined as

$$d = \sum_{t=2}^{n} \left( e_t - e_{t-1} \right)^2 \bigg/ \sum_{t=1}^{n} e_t^2 \tag{12}$$

where $n$ is the number of time step. The value of $d$ ranges from 0 to 4, with $d = 2$ indicating no autocorrelation. A value of $d < 2$ suggests positive autocorrelation, while $d > 2$ indicates negative autocorrelation. In our analysis, most Durbin–Watson statistics were close to 2, and there was no systematic deviation toward either positive or negative autocorrelation (Supplementary Figs. 4b, 5b and 9b), indicating little to no autocorrelation overall.

### Random Forest regression analysis

We applied the RF algorithm to perform both interannual and trend analyses, based on the independent and dependent variables defined in Eqs. (2)–(6). Prior to conducting the interannual analysis, long-term linear trends were removed from all variables. The detrended time series were then converted into z-scores by dividing by the standard deviation of the anomalies. For the interannual analysis, the RF regression model was combined with the SHAP[36,88] framework to quantify the relative importance of various environmental variables in driving the interannual variability of VPD, SVP, AVP, and RH. The SHAP method, grounded in the Shapley value concept from cooperative game theory, offers an enhanced approach over traditional local interpretable model-agnostic explanations by providing consistent and theoretically sound attributions of feature importance. For each response variable $Y$ (e.g., VPD, SVP, etc.) and each sample $i$, the RF-predicted outcome can be decomposed as:

$$Y_i = Y_{base} + \sum_{j=1}^{M} shap\left(x_{ij}\right) \qquad (13)$$

where $Y_i$ is the RF prediction for sample $i$, $Y_{base}$ is the mean prediction across all samples (i.e., the expected value of the model output), and $shap(x_{ij})$ is the SHAP value representing the contribution of predictor $j$ to the prediction $Y_i$, and $M$ is the number of predictor variables.

The relative importance of each predictor variable was quantified using the normalized magnitude of its absolute SHAP values across all samples. Specifically, for predictor $j$, the relative importance $RI_j$ was calculated as:

$$RI_j = \frac{\frac{1}{N}\sum_{i=1}^{N}\left|shap\left(x_{ij}\right)\right|}{\sum_{k=1}^{M}\frac{1}{N}\sum_{i=1}^{N}\left|shap\left(x_{ij}\right)\right|} \qquad (14)$$

where $N$ is the total number of samples. The vertical bars denote absolute value. This metric captures the average absolute contribution of each variable to the model output, normalized across all predictors, and serves as a measure of overall variable importance. A predictor is considered the dominant driver if it shows the highest mean absolute SHAP value among all predictors, indicating that it contributes the most to the RF regression model output.

To determine whether each variable had a positive or negative influence on the response variable, we calculated the Theil–Sen slope between the values of each predictor and its corresponding SHAP values. A positive slope indicates that increases in the predictor tend to increase its SHAP value contribution (i.e., positive influence on $Y$), while a negative slope indicates that increases in the predictor tend to decrease its SHAP value contribution (i.e., negative influence on $Y$). This approach allows us to assess not only which variables are most important in driving model predictions, but also whether their effects are positive or negative.

For the trend analysis, we used all monthly data without detrending to train the RF model to predict VPD, SVP, and RH, as defined in Eqs. (2), (4), and (6), respectively. Model performance was evaluated using the out-of-bag (OOB) coefficient of determination ($R^2$) and mean squared error (MSE), which are internal cross-validation procedure inherent to RF and provide unbiased estimates of predictive accuracy without requiring a separate validation dataset. The long-

term impacts of sub-diurnal asymmetric warming on changes in VPD, SVP, and RH were then quantified using the RF model combined with a series of sensitivity experiments, as described below.

All RF models were trained individually at each station or grid point using 100 decision trees. Each tree independently predicted the response variable based on the provided predictors. To determine the optimal number of trees, we conducted a sensitivity analysis (Supplementary Fig. 28) using monthly variables from HadISD observations and the model defined in Eq. (2). The results show that the OOB MSE stabilizes once the number of trees exceeds 30, indicating that using 100 trees is sufficient for our purposes.

### Sensitivity experiments for trend analysis

To assess the relative contribution of individual predictors (e.g., temperature and SM) to changes in a target variable (e.g., VPD or RH), we conducted a series of sensitivity experiments[9] based on the RF regression models. Let $Y$ denote the target variable (e.g., VPD or RH), and let $X = (X_1, X_2,..., X_n)$ represent the set of input predictors (e.g., $T_{max}$, $T_{min}$, SM). For each RF regression model, we first trained the model using the full dataset (1980–2023) to optimize its parameters, then applied the trained model on the input data to obtain fitted $Y$ values ($Y_{fitted}$):

$$Y_{fitted} = f(X_1, X_2, \ldots, X_n) \qquad (15)$$

To quantify the contribution of a single predictor $X_i$ to the change in $Y$, we performed sensitivity experiments by fixing $X_i$ at its climatological monthly mean during a control period (1980–1982), while allowing all other variables to vary as observed. A new prediction $Y_{-Xi}$ was made under this perturbed condition:

$$Y_{\sim Xi} = f(X_1, \ldots X_{i,ctrl}, \ldots, X_n) \qquad (16)$$

where $X_{i,ctrl}$ is set to the multi-year mean of $X_i$ for the corresponding calendar month calculated over the control period (1980–1982) and held constant throughout the time series. The contribution of $X_i$ to the change in $Y$, denoted as $Y_{Xi}$, was then calculated as:

$$Y_{Xi} = Y_{fitted} - Y_{\sim Xi} \qquad (17)$$

This process was repeated for each predictor, resulting in separate estimates of the contributions from each variable. This framework allowed us to isolate and quantify the role of each variable in driving long-term changes in the target variable.

### Seasonal analysis and detrending for seasonality

In the analysis of FLUXNET site observations, seasons were defined as follows: March, April, May for spring (autumn), June, July, August for summer (winter), September, October, November for autumn (spring), and December, January, February for winter (summer) in the Northern Hemisphere (Southern Hemisphere).

To remove the seasonal cycle from the original daily data, we utilized a Fourier transform approach. First, we computed the mean daily values across all years and then applied a Fast Fourier Transform (FFT)[54] to these mean values. To retain only the primary seasonal components, we filtered the frequencies by zeroing out higher frequencies while preserving only the four lowest frequency components. We then performed an inverse FFT on the filtered data to reconstruct the seasonal component in the time domain. Finally, we subtracted the seasonal component from the original daily data to obtain the deseasonalized data. To maintain consistent 365-day years, we excluded the last day of the year during leap years.

## Decoupling of SWC and DTR or VPD

Based on the daily observations at FLUXNET sites with seasonality removed, we calculated the SWC percentiles (4th through 96th, at 4-percentile intervals) at each tower site. These percentiles values were then used to bin the data. Data for all variables (temperature, SWC, VPD, etc.) were sorted into 25 bins according to the percentiles of SWC. This binning procedure maintained the temporal match between data points. Only bins with more than 40 data points were included in the further analysis. Since SWC is largely decoupled from DTR and VPD within each SWC bin (Supplementary Fig. 15a), this method mitigates the influence of soil moisture on the impact of diurnal temperature variation on VPD.

## Data availability

All data needed to evaluate the conclusions in the paper are present in the paper and/or the Supplementary Materials. The source data underlying Figs. 1–4 have been deposited in the Figshare repository and are available at https://doi.org/10.6084/m9.figshare.29940365.v1. The HadISD dataset is from https://www.metoffice.gov.uk/hadobs/hadisd/. The ERA5-Land dataset is from https://cds.climate.copernicus.eu/datasets/reanalysis-era5-land?tab=download. The FLUXNET2015 dataset is from https://fluxnet.org/data/fluxnet2015-dataset/. The scPDSI data is from https://crudata.uea.ac.uk/cru/data/drought/#global. The FWI data is from https://ewds.climate.copernicus.eu/datasets/cems-fire-historical-v1.

## Code availability

The code for the analysis and mapping can be obtained from the Figshare repository (https://doi.org/10.6084/m9.figshare.29940365.v1).

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

## Acknowledgements

The study was supported by internal funding from Chalmers University of Technology. Z.Z was supported by the VAPOR project (grant number 101154385), funded by the Horizon Europe, MSCA Postdoctoral Fellowships 2023. A.D. acknowledges the support of the National Science Foundation (grant number AGS-2015780).

## Author contributions

Z.Z. designed the research, performed the analysis and wrote the draft; H.W.C., A.D., T.Z., B.H. and B.S. provided comments to improve the manuscript; H.W.C. supervised the project.

## Funding

## Competing interests

The authors declare no competing interests.
