## [Transparent Peer Review file · Nature Communications]

Sub-diurnal asymmetric warming has amplified atmospheric dryness since the 1980s

Corresponding Author: Dr Hans Chen

Version 0:

Reviewer comments:

Reviewer #1

(Remarks to the Author)

General Comments:

The authors quantified the relative importance of Tmax and Tmin in driving VPD variations based on site observations and reanalysis data. They then used a random forest model to quantify the impact of DTR (Tmax – Tmin) changes including trends, on VPD, SVP, and RH. However, the study needs to address the following main considerations:

Robustness of Ridge Regression Analysis: The authors applied ridge regression to quantify the relative importance of Tmax and Tmin in interannual VPD variations after removing trends. While ridge regression accounts for multicollinearity to some extent, the analyzed atmospheric covariates exhibit strong interactions. The authors need to consider additional approaches to ensure the robustness of their results.

Issues with Random Forest Regression: The manuscript consistently refers to 'diurnally asymmetric warming', but does DTR exhibit a uniform increasing trend over land? Apart from Fig. 1a, please provide a spatial distribution map of DTR trends, along with specific statistics:

What proportion of the land area shows an increasing DTR trend?

What proportion exhibits a decreasing DTR trend?

The term 'diurnally asymmetric warming' may be misleading; I suggested using a more neutral expression.

Regarding the random forest simulations, more details are needed, especially about the experimental design, to ensure the reproducibility of the methodology and corresponding results.

Specific Comments:

Line 21-24: Do you mean "Faster increases in Tmax play a dominant role in rising VPD compared to Tmin"?

Line 55-56: This sentence is difficult to understand. Do you mean that previously faster nighttime warming has now shifted to faster daytime warming?

Line 56-57: If my understanding is correct, you are examining the relative importance of Tmax and Tmin warming in driving VPD variations.

Line 66: Please consistently define the significance criterion throughout the manuscript. Are you using $p < 0.05$, which is commonly applied, or $p < 0.1$, which is sometimes used in climate change research?

Line 78-80: The variable SM suddenly appears here. Does it directly influence the relationship between DTR and VPD? The authors explain that SM is relevant due to its role in evaporative cooling, but why not use evapotranspiration instead?

Line 82: The five variables (SVP, Tmin, Tmax, DTR, and SM) are highly correlated. The authors should consider multicollinearity issues and factor interaction effects. I suggest incorporating machine learning techniques to further ensure the robustness of the results.

Figure 1:

While the title is "Diurnally Asymmetric Warming and Its Impact on Vapor Pressure Deficit in 1980–2023", the core focus is actually on the relative importance of Tmax and Tmin in interannual VPD variations.

The caption of Fig. 1 needs to be revised accordingly. Since the authors conducted ridge regression using detrended time series, Panel A might be misleading. It could serve as background information but may be more appropriate as supplementary material.

The ridge regression results represent regression coefficients of the dependent variable (Y) to independent variables (X). The authors should clearly explain this concept in advance.

The relative importance of Tmax and Tmin is assessed by comparing regression coefficients, so I suggest using "relative importance" or "relative dominance" instead of "contribution", which might be ambiguous.

The exact meaning of R.c should be explicitly explained rather than referring to the Methods section, as this increases the difficulty of understanding.

To enhance clarity, I recommend using the full term "regression coefficients" in the figure instead of 'R.c'.

Did the authors normalize the time series before regression? If not, the differences in variable magnitudes could affect the results.

Since regression coefficients include both positive and negative values, I suggest adding explicit statistics in Panels B, C, and E to indicate the proportion of positive vs. negative coefficients across all grid cells.

Lines 110-112: This sentence should be rewritten for clarity. Consider:

"Given that VPD is calculated as $VPD = SVP - AVP$, and $AVP = SVP * (1 - RH)$, we conducted separate ridge regression analyses ..."

Lines 110-136: When conducting ridge regression for SVP, was SM included as an independent variable? Please clarify this.

Lines 131-133: Are you sure? In arid regions, the mean annual RH can be ~60%. This statement seems inconsistent with observations.

Figure 2:

Since SVP is computed using an exponential function of temperature, why is SM included as an independent variable? This is counterintuitive.

Please add Tmax, Tmin, and SM as X-axis variables in each sub-panel.

Similar to Fig. 1, please provide explicit statistics showing the proportion of positive vs. negative regression coefficients across all grid cells.

Figure 4: For Panels D-G, please provide explicit statistics indicating the proportion of positive vs. negative trends across all grid cells.

Lines 303-315:

Regarding VPD calculation, I suggest referencing Allen et al. (1998) and explicitly listing the exact equation (see pages 37–39 of their paper).

Did the site observations provide dew point temperature (Tdew)?

If so, I strongly recommend using Tdew to compute actual vapor pressure because:

Tdew directly represents atmospheric moisture content.

RH estimates in reanalysis datasets contain significant uncertainties, particularly in trend analysis.

Reference:

Allen, R.G., Pereira, L.S., Raes, D., and Smith, M. (1998). Crop Evapotranspiration: Guidelines for Computing Crop Water Requirements (FAO).

Lines 316-336:

Were variables standardized to eliminate dimensional influence?

Besides multicollinearity, the analyzed atmospheric variables exhibit strong interactions.

I question whether the current approach sufficiently addresses these interaction effects and ensures the robustness of the results.

Lines 364-378:

The study employs a random forest model with 100 decision trees, but why was 100 chosen?

Was a sensitivity analysis conducted to determine an optimal tree count?

The manuscript states that the dataset was split (80% training, 20% validation), followed by a final retraining on the entire dataset.

What was the purpose of this final retraining? Was it to improve model stability, generate final predictions, or something else?

Sensitivity Experiments: The description lacks sufficient detail. Were individual variables perturbed while others remained constant? Were fixed increments (e.g., 1%, 2%, 5%) applied, or were variations confined to a specific statistical range (e.g., one standard deviation)? The formula or approach for computing variable contributions should be explicitly stated.

To enhance clarity and reproducibility, the Methods section should provide a structured and detailed explanation of each analytical step, ensuring that all procedures can be independently replicated.

Reviewer #2

(Remarks to the Author)

Review of manuscript NCOMMS-25-04256-T

Diurnally asymmetric warming has amplified atmospheric dryness since the 1980s

Ziqian Zhong, Hans W. Chen, Aiguo Dai, Tianjun Zhou, Bin He, Bo Su

Overview

This study examines the increase in atmospheric vapour pressure deficit since the 1980s and attempts to establish the contribution to this increase from diurnally asymmetric trends in daily maximum and daily minimum temperatures. The study uses sub-daily data from weather stations and ERA5 reanalysis, applying regression methods to characterise the relationships between atmospheric dryness (vapour pressure deficit), near surface temperature and moisture. The principal finding is that the faster rise in daily maximum temperatures compared to daily minimum temperatures over land has intensified atmospheric dryness from 1980 to 2023.

Overall, this manuscript addresses an important topic and the findings are noteworthy. However, there are areas where the

manuscript could be strengthened by major revisions. My comments and suggestions are listed below.

General comments

1. The results prompt important questions that merit further exploration, such as “Did VPD decline during 1950-1980 and was it driven by diurnally asymmetric temperature trends (which had the opposite sign to the post 1980 period)?” If the analysis is not extended, then I suggest including discussion of pre 1980 trends and the potential applicability of these results to the pre 1980 trends. The results of the paper would be more compelling if the period prior to 1980 is included in the analysis.
2. The comparison of ERA5 reanalysis and HADISD stations. I think it important to compare results from these two sources on a consistent basis. I suggest a secondary result is calculated using a subset of ERA5 grid cells which are co-located with the stations. This would enable discussion of the differences. Further, inferences could be made as to the causes of the differences. For instance, when I have done this in the past, I have found that differences in altitude between the stations and the paired ERA5 grid cells can explain a reasonable proportion of the difference in temperatures. The distribution of the stations is strongly skewed towards the USA and Europe, comment could be made about differences between stations and ERA5 in different regions. These changes would enhance confidence in the ERA5-based results and provide a robust validation framework.
3. Regression models. I would like to see more results on the quality of the fit of the models (mainly in SI) and I think it essential that the limitations of the models are discussed in the Conclusion section. Linear regression models are used while the relationship between temperature and SVP is strongly non-linear. How does this affect the results?
4. The Conclusion section could be improved with a critical assessment of the study findings versus recent literature and a more in-depth discussion of the relevance of these results for climate impacts such as drought and wildfires. This would underscore the relevance of the findings and their broader implications and, in my opinion, is particularly relevant for a journal such as Nature Communications.
5. The impact of the study would be improved if results are presented in a way which quantitatively connects regional atmospheric drying (and its drivers) with regional climate impacts such as drought and wildfires. For instance, trends and regression results could be summarised for key regions of drought and escalated wildfire occurrence during 1980-2020s.

Detailed comments

6. Lines 18-19. I assume the VPD, SVP, and AVP are defined/measured at the near surface (~2m). It would make the text clearer to explicitly state the level at which these terms are defined.
7. Lines 28-29 plus all figures and results in the paper which present results based on ERA5 reanalysis and station observations. This statement, and the presentation of results more generally in the paper, compares results from station observations and ERA5 for different regions. Therefore, it does not serve to validate the use of the ERA5 reanalysis. To validate the results based on ERA5 against observations, I suggest pairing each station with its nearest neighbour grid cell from ERA5 and presenting the results for this subset of grid cells.
8. Line 30 and other places which refer to drivers of the changes. The study would be strengthened by unpacking “global warming”. For instance, it is important to mention changes in aerosols. Trends in aerosol emissions have been key to the observed changes in diurnal temperature range (Wild 2009, Zhou et al., 2010).
9. Lines 58-61. Results from many regression models are presented in the main paper and in supplementary information. It might help the reader to outline the different methods and results in this paragraph. This would help by outlining the structure of the paper and provide a high-level guide to the how the research question is addressed.
10. Line 65 and other places where formulae are disclosed. It would help the reader to define the equations and reference them with numbers throughout. This will avoid repeating the same formula through the manuscript.
11. Line 69. I don't understand the phrase “large-scale characteristics of the HadISD data”. This dataset is a collection of point measurements. Are you saying that the discontinuities in hydrometer measures are localised and small in number? It would help clarity to be more specific in the text.
12. Lines 70-71. Are you referring to HadISD data or ERA5 reanalysis? I suggest you review the manuscript text to ensure it is clear throughout which results are referred to.
13. Lines 78-79 and use of regression methods generally. It could strengthen the robustness of the study to explore the assumptions of the regression models in greater detail. For instance, you could comment on autocorrelation and the distribution of residuals.
14. Line 88. It would help clarity in the main text to explain the measure you use to quantify contribution.
15. Figure 1 (and other figures too). It would help clarity to have descriptive panel titles within the figures.
16. Line 97. It took me in a few moments to understand the following: “Spatial distribution of ridge regression coefficients (R.c., see Methods) of VPD with respect to DTR, with DTR, soil moisture (SM), and mean air temperature as independent variables.” I think it would be clearer to define an equation for each regression in the text and cross-reference the equations here and elsewhere.
17. Line 106. I cannot see where the non-significant grid cells and stations are located. Have they been hatched or masked in a specific colour? Further, it would be informative to quote the proportion of stations and grid cells which are significant in each panel. I have the same comment for Figure 2 (lines 148-149).
18. Figure 2. It could help interpretation to change the colour scale. Differences between regional results would be clearer. Results located in the 3 vertices of the triangle are clear enough, but colours towards the centre of the triangle are not so clear. I suggest you try using colours for discrete categories (e.g., no significant contribution from Tmx, Tmn, sm; Tmx only; Tmn only; etc) rather than a continuous colour scale.
19. Line 215. I assume that the validation set is the remaining 20% - I suggest you explicitly say this to help clarity.
20. Line 272-274. This sentence requires a more nuanced discussion. Mention could be made of: trends prior to 1980, actual projections from CMIP6, the drivers of diurnal asymmetry in warming (particularly regional changes in aerosols and solar dimming/brightening).
21. Line 280. State explicitly the time frequency of the data used.
22. Line 290. Could you state explicitly the depth of the layer used?
23. Line 300. Why did you use soil moisture between 0-1m depth? It appears inconsistent with the topmost layer used from

Fluxnet (which I assume is shallower than 1 m depth).

24. Line 327: Please justify use of the threshold of 3.

25. Line 370. What metrics did you use to validate the random forest? I would expect to see several metrics such as RMSE, MSE, area under curve (AUC), out-of-bag (OOB). Key results from this in SI would demonstrate the robustness of the study.

26. Paragraph at line 372. What quantitative techniques were used to interpret and explain the random forest models? For instance, did you use SHAP values?

27. References – You could consider citing Zhou et al. (2010) <https://link.springer.com/article/10.1007/s00382-009-0644-2>

28. SI Figure 1 caption. I found the statement “over land areas during” confusing because the station observations are point measurements and are not representative of large-scale land areas.

29. SI Figure 2. I would expect to see statistical significant tests applied to the gridded maps. This comment applies to all maps in the paper.

30. SI Figure 2. This figure sets up the research question very nicely. You might consider moving it to be the first figure in the main paper. Panel c) shows that trends in DTR vary between regions. This is relevant to the study and something that could be commented on in the main text.

31. SI Figure 2. Should 1980-2022 be 1980-2023?

32. SI Figure 3. “Only the stations or grids with the regression result that passed the test of significance”. I can’t see where the excluded grid cells are. How are they shown (e.g., masked out or hatched?). It would be informative to state the proportion of stations & grid cells that are significant.

References

Wild (2009) <https://agupubs.onlinelibrary.wiley.com/doi/10.1029/2008JD011470>

Zhou et al. (2010) <https://link.springer.com/article/10.1007/s00382-009-0644-2>

Version 1:

Reviewer comments:

Reviewer #1

(Remarks to the Author)

Thank you for addressing my previous comments. The current version shows substantial improvement. Please consider the following minor comments below:

Line 63-65: To improve clarity and precision, consider revising lines 63-65 to: "Given the potential complex interactions and non-linear relationships among the examined climatic factors, we utilized Random Forest modeling as a complementary approach." Additionally, include one or two sentences such as: "This approach delivers robust predictive performance by effectively capturing intricate, non-linear data patterns that traditional linear models may not adequately address.

Furthermore, Random Forest modeling provides feature importance rankings, enabling a clearer understanding of each climatic factor's relative contribution." Please note that I am striving to align the writing more closely with addressing the research challenges, so please adjust as appropriate based on the specifics of the research work.

Line 76-77: Please add a reference for the calculation of vapor pressure deficit (VPD). The method described in Allen et al. (1998) is widely cited and would be appropriate to include.

Allen, R. G., Pereira, L. S., Raes, D., & Smith, M., 1998. Crop evapotranspiration Guidelines for computing crop water requirements, FAO Irrigation and Drainage Paper 56, Food and Agriculture Organization of the United Nations.

Line 78-96: I suggest separating this section into two paragraphs: one focusing on the explanation of VPD trends and the other on DTR trends.

Line 118-122: Eqn (2) and Eqn (3): The current formulation of these equations is not rigorous. As written, they imply that the authors aim to predict VPD, whereas the actual purpose appears to be assessing the relative importance of the explanatory variables. The authors might consider using a notation such as “ $VPD \sim f(DTR, T_m, S_M)$ ” and “ $VPD \sim f(T_{max}, T_{min}, S_M)$ ” to clarify that VPD is simply the response variable rather than a forecast target. Given that the relationships among variables are already explained in Lines 118–120, and the number of variables is relatively small, the authors could also consider omitting the equation expressions. In addition, please use ‘Tmean’ instead of ‘TM’ throughout the manuscript to ensure consistency with the notation for Tmin and Tmax.

Line 142-146: Similarly, if the research objective can be clearly articulated in the text, it may not be necessary to present the equations here, as they are more appropriate for the Methods section.

Line 204: ‘Fig. 2b-2c?’

Line 490: Please clarify the proportion of data used for training and testing. Additionally, did the authors perform cross-validation? If so, please specify the number of folds used.

Fig. 1: Please note that some text elements in the figure are overlapping. Make sure the figure is clear and fully legible.

Reviewer #2

(Remarks to the Author)

Review of manuscript NCOMMS-25-04256A

Sub-diurnal asymmetric warming has amplified atmospheric dryness since the 1980s
Ziqian Zhong, Hans W. Chen, Aiguo Dai, Tianjun Zhou, Bin He, Bo Su

I am happy that my comments have been addressed and find the manuscript substantially improved. I have no further comments to make on the manuscript.

Response to reviewer 1

The authors quantified the relative importance of Tmax and Tmin in driving VPD variations based on site observations and reanalysis data. They then used a random forest model to quantify the impact of DTR (Tmax – Tmin) changes including trends, on VPD, SVP, and RH.

Res: Thank you very much for your constructive comments and positive evaluation of our manuscript. Please find our point-by-point responses below addressing the comments you suggested.

However, the study needs to address the following main considerations:

Robustness of Ridge Regression Analysis: The authors applied ridge regression to quantify the relative importance of Tmax and Tmin in interannual VPD variations after removing trends. While ridge regression accounts for multicollinearity to some extent, the analyzed atmospheric covariates exhibit strong interactions. The authors need to consider additional approaches to ensure the robustness of their results.

Res: Thank you very much for your constructive comment. In the revised manuscript, in addition to the ridge regression analysis, we have applied a Random Forest model with Shapley Additive Explanations (RF-SHAP) framework to quantify the relative importance of the different environmental variables on interannual VPD variations. This method helps mitigate the effects of strong interactions and multicollinearity among independent variables, thereby enhancing the robustness of our results.

Issues with Random Forest Regression: The manuscript consistently refers to ‘diurnally asymmetric warming’, but does DTR exhibit a uniform increasing trend over land? Apart from Fig. 1a, please provide a spatial distribution map of DTR trends, along with specific statistics:

What proportion of the land area shows an increasing DTR trend?

What proportion exhibits a decreasing DTR trend?

Res: DTR does not exhibit a uniform increasing trend over land. However, our analysis shows that most (58%) of the global land area experienced an increasing DTR trend, while 42% exhibited a decreasing trend during the period 1980–2023, as presented in the revised Figure 1a. This proportion of increasing DTR is consistent with findings from our previous research (Zhong et al., 2023), which reported that the percentage of land area with increasing DTR during a similar period (1991–2020) ranged from 52% based on the Climatic Research Unit Time-Series (CRU-TS) to 70% based on Berkeley Earth Surface Temperatures (BEST) dataset.

References

Zhong, Z., He, B., Chen, H.W., Chen, D., Zhou, T., Dong, W., Xiao, C., Xie, S.-p., Song, X., Guo, L., Ding, R., Zhang, L., Huang, L., Yuan, W., Hao, X., Ji, D., & Zhao, X. (2023). Reversed asymmetric warming of sub-diurnal temperature over land during recent decades. *Nature Communications*, 14, 7189

The term ‘diurnally asymmetric warming’ may be misleading; I suggested using a more neutral expression.

Res: To maintain clarity and neutrality, we have revised the manuscript to use a more descriptive and neutral expression—"sub-diurnal asymmetric warming"—throughout the text.

Regarding the random forest simulations, more details are needed, especially about the experimental design, to ensure the reproducibility of the methodology and corresponding results.

Res: We have added more detailed descriptions of the random forest simulations in the Methods section to ensure the reproducibility of our methodology and corresponding results. Please see lines 490–553.

Specific Comments:

Line 21-24: Do you mean "Faster increases in Tmax play a dominant role in rising VPD compared to Tmin"?

Res: Yes. We have revised the corresponding expression in the manuscript for clarity.

Line 55-56: This sentence is difficult to understand. Do you mean that previously faster nighttime warming has now shifted to faster daytime warming?

Res: Yes, that is what we intended to convey. We have revised the sentence based on your suggestion to improve clarity.

Line 56-57: If my understanding is correct, you are examining the relative importance of Tmax and Tmin warming in driving VPD variations.

Res: Correct. We have revised the sentence accordingly to clarify this point.

Line 66: Please consistently define the significance criterion throughout the manuscript. Are you using $p < 0.05$, which is commonly applied, or $p < 0.1$, which is sometimes used in climate change research?

Res: We have consistently defined the significance criterion as $p < 0.05$ throughout the revised manuscript.

Line 78-80: The variable SM suddenly appears here. Does it directly influence the relationship between DTR and VPD? The authors explain that SM is relevant due to its role in evaporative cooling, but why not use evapotranspiration instead?

Res: SM does not directly influence the relationship between DTR and VPD. However, it is essential to control for the effect of SM when analyzing this relationship. This is because SM is strongly negatively correlated with both DTR and VPD. As a result, the observed positive relationship between DTR and VPD may, in part, arise from their

mutual negative dependence on SM rather than from a direct causal link. Therefore, we included SM as an independent variable in the regression model to mitigate its confounding effect and isolate the direct relationship between DTR and VPD. Please see line 123-127.

We agree that, in theory, evapotranspiration (ET) would be a more direct indicator due to its role in surface energy and water fluxes. However, ET is inherently difficult to measure accurately because it is influenced by a complex combination of environmental and biophysical factors. For this reason, we used SM as a more reliable proxy in the main analysis. Nonetheless, to enhance the robustness of our results, we also performed additional regression analyses using ET estimates as an independent variable. These discussions are now included in the revised manuscript, please see line 357-370 for more details.

Line 82: The five variables (SVP, Tmin, Tmax, DTR, and SM) are highly correlated. The authors should consider multicollinearity issues and factor interaction effects. I suggest incorporating machine learning techniques to further ensure the robustness of the results.

Res: We have incorporated a Random Forest model combined with SHAP framework to quantify the relative importance of different environmental variables on interannual VPD variations. This RF-SHAP framework helps to mitigate multicollinearity issues and capture complex interaction effects among variables. Please refer to Figs. 1 and 2 in the revised manuscript for the updated analysis.

Figure 1: While the title is "Diurnally Asymmetric Warming and Its Impact on Vapor Pressure Deficit in 1980–2023", the core focus is actually on the relative importance of Tmax and Tmin in interannual VPD variations. The caption of Fig. 1 needs to be revised accordingly.

Res: Good suggestion. We have revised the caption of Fig.1 accordingly.

Since the authors conducted ridge regression using detrended time series, Panel A might be misleading. It could serve as background information but may be more appropriate as supplementary material.

Res: We agree that the original Panel A could be misleading and is more appropriate for the supplementary material. Combined with your and another reviewer's suggestions, we have replaced this figure in the revised manuscript with the spatial distribution of DTR trends, which was originally included in the supplementary material.

The ridge regression results represent regression coefficients of the dependent variable (Y) to independent variables (X). The authors should clearly explain this concept in advance.

Res: Thank you for your comment. In response to your suggestion, as well as similar feedback from another reviewer, we have clearly defined the regression equations for each model in advance. Please refer to the revised caption of Figs. 1 and 2 for details.

The relative importance of Tmax and Tmin is assessed by comparing regression coefficients, so I suggest using "relative importance" or "relative dominance" instead of "contribution", which might be ambiguous.

Res: We have replaced the term "contribution" with "relative importance" in the revised manuscript to avoid ambiguity.

The exact meaning of R.c should be explicitly explained rather than referring to the Methods section, as this increases the difficulty of understanding. To enhance clarity, I recommend using the full term "regression coefficients" in the figure instead of 'R.c'.

Res: In the revised figure captions of Figs. 1 and 2, we have explicitly clarified the meaning of the regression coefficients, including which ridge regression model they

refer to and the specific variables involved. To enhance clarity, we have also replaced the abbreviation “R.c” with the full term “Regression coefficient” in the figures.

Did the authors normalize the time series before regression? If not, the differences in variable magnitudes could affect the results.

Res: Yes; prior to conducting the regression analysis in the section “Interannual analysis”, long-term linear trends were removed from all variables. The resulting detrended time series were then normalized by converting them into z-scores, calculated as anomalies relative to the linear trend divided by the standard deviation over the period 1980–2023. Please see the revised captions of Figs. 1 and 2 and line 457-459.

Since regression coefficients include both positive and negative values, I suggest adding explicit statistics in Panels B, C, and E to indicate the proportion of positive vs. negative coefficients across all grid cells.

Res: We have added pie chart insets to show the land area percentage with positive versus negative regression coefficients based on ERA5-Land data. Please see the revised Panels B, C, and E of Fig. 1.

Lines 110-112: This sentence should be rewritten for clarity. Consider:

*"Given that VPD is calculated as $VPD = SVP - AVP$, and $AVP = SVP * (1 - RH)$, we conducted separate ridge regression analyses ..."*

Res: Thank you for your suggestion. For clarity, we have revised the sentence. Please see the updated version in lines 142–143 of the revised manuscript.

Lines 110-136: When conducting ridge regression for SVP, was SM included as an independent variable? Please clarify this.

Res: In the revised manuscript, SM was removed and not included as an independent variable in the ridge regression analysis for SVP.

Lines 131-133: *Are you sure? In arid regions, the mean annual RH can be ~60%. This statement seems inconsistent with observations.*

Res: We agree that RH is relatively low in arid regions, and the annual mean value of ~60% you mentioned is indeed much lower than 100%. However, hourly RH can still approach 100% even in arid environments. One line of evidence is the formation of dew, which serves as an important water source for plants in semi-arid and arid regions (Zhang et al., 2019), including deserts (Jia et al., 2014; Yu et al., 2020). Dew formation requires the surface temperature to fall to or below the dew point, at which RH approaches 100%. Nonetheless, we have revised the sentence in the manuscript to make it more rigorous: “Typically, over land—especially under more humid conditions—RH often approaches 100% around the time of the Tmin, and AVP (or specific humidity) is relatively stable throughout the 24-hour diurnal cycle ...”.

References

- Jia, R.-l., Li, X.-r., Liu, L.-c., Pan, Y.-x., Gao, Y.-h., & Wei, Y.-p. (2014). Effects of sand burial on dew deposition on moss soil crust in a revegetated area of the Tennger Desert, Northern China. *Journal of Hydrology*, 519, 2341-2349
- Yu, R., Zhang, Z., Lu, X., Chang, I.S., & Liu, T. (2020). Variations in dew moisture regimes in desert ecosystems and their influencing factors. *WIREs Water*, 7, e1482
- Zhang, Y., Hao, X., Sun, H., Hua, D., & Qin, J. (2019). How *Populus euphratica* utilizes dew in an extremely arid region. *Plant and Soil*, 443, 493-508

Figure 2:

Since SVP is computed using an exponential function of temperature, why is SM included as an independent variable? This is counterintuitive.

Res: We agree that including SM here is unnecessary and counterintuitive. In the

revised manuscript, only Tmax and Tmin are used as independent variables in the regression model when SVP is the dependent variable, please see Equation (4) in the revised manuscript.

Please add Tmax, Tmin, and SM as X-axis variables in each sub-panel.

Res: Thank you for pointing this out. Following another reviewer's suggestion, we have replaced the figure. We believe the revised version now provides the necessary information. Please refer to the updated Fig. 2 in the revised manuscript.

Similar to Fig. 1, please provide explicit statistics showing the proportion of positive vs. negative regression coefficients across all grid cells.

Res: We have added pie chart insets to illustrate the land area percentages with positive and negative regression coefficients, please see the revised Figs. 1 and 2.

Figure 4: For Panels D-G, please provide explicit statistics indicating the proportion of positive vs. negative trends across all grid cells.

Res: This statistical information has been added as pie charts in the revised Fig. 4.

Lines 303-315:

Regarding VPD calculation, I suggest referencing Allen et al. (1998) and explicitly listing the exact equation (see pages 37–39 of their paper).

Res: Thank you for your suggestion. The VPD calculation equation referenced in Allen et al. (1998) is based on Tetens' formula for temperatures above 0 °C, which is mathematically equivalent to the equation we list in the manuscript for air temperatures ≥ 0 °C (with constants $a = 17.269$ and $b = 237.3$). This formulation is indeed appropriate for calculating monthly VPD, as done in Allen et al. (1998). However, when applied to hourly data—especially under extremely cold conditions—this formulation can

introduce substantial uncertainty. Previous studies have shown that in cold, high-latitude regions, the VPD estimation error increases as temperature decreases (Junzeng et al., 2012). Therefore, to improve accuracy under such conditions, we adopt Tetens' formulation for temperatures below 0 °C as recommended by Murray (1967).

References

- Junzeng, X.U., Qi, W.E.I., Shizhang, P., & Yanmei, Y.U. (2012). Error of Saturation Vapor Pressure Calculated by Different Formulas and Its Effect on Calculation of Reference Evapotranspiration in High Latitude Cold Region. *Procedia Engineering*, 28, 43-48
- Murray, F.W. (1967). On the Computation of Saturation Vapor Pressure. *Journal of Applied Meteorology and Climatology*, 6, 203-204

Did the site observations provide dew point temperature (Tdew)? If so, I strongly recommend using Tdew to compute actual vapor pressure because: Tdew directly represents atmospheric moisture content. RH estimates in reanalysis datasets contain significant uncertainties, particularly in trend analysis.

Res: Yes, the site observations provide Tdew. Following your suggestion, we calculated VPD based on Tdew in the revised manuscript. The main conclusions remain unchanged.

Were variables standardized to eliminate dimensional influence?

Res: Yes. All variables were standardized using z-scores prior to the interannual analysis to eliminate the influence of differing units and magnitudes. Please see the updated version in lines 457-459 of the revised manuscript.

Besides multicollinearity, the analyzed atmospheric variables exhibit strong interactions. I question whether the current approach sufficiently addresses these interaction effects and ensures the robustness of the results.

Res: In the revised manuscript, we incorporated additional analyses using a RF-SHAP

framework to evaluate the relative importance of the different environmental variables on interannual VPD variations. The SHAP-based importance metrics (i.e., mean absolute SHAP values) used in this study reflect the overall contribution of each predictor to the model output, incorporating both main effects and interactions. This approach provides a robust and model-agnostic assessment of variable influences in complex, non-linear models like Random Forests. This framework effectively mitigates the influence of multicollinearity and accounts for interaction effects, thereby enhancing the robustness of the results.

Lines 364-378: The study employs a random forest model with 100 decision trees, but why was 100 chosen? Was a sensitivity analysis conducted to determine an optimal tree count?

Res: We conducted a sensitivity analysis to determine the optimal number of decision trees, as shown in Supplementary Fig. 28. The result shows that the out-of-bag (OOB) mean squared error (MSE) of the Random Forest model stabilizes when the number of trees exceeds 30. This indicates that using 100 decision trees is sufficient for the purposes of our analysis.

The manuscript states that the dataset was split (80% training, 20% validation), followed by a final retraining on the entire dataset. What was the purpose of this final retraining? Was it to improve model stability, generate final predictions, or something else?

Res: Thank you for your comment. In the revised manuscript, we improved the methodology by training the RF model using the full dataset and evaluating its performance using out-of-bag (OOB) R^2 and OOB MSE. This approach ensures that model evaluation remains unbiased without the need for a separate validation set, and the OOB metrics provide a robust estimate of model performance.

Sensitivity Experiments: The description lacks sufficient detail. Were individual

variables perturbed while others remained constant? Were fixed increments (e.g., 1%, 2%, 5%) applied, or were variations confined to a specific statistical range (e.g., one standard deviation)? The formula or approach for computing variable contributions should be explicitly stated.

Res: In our analysis, we used monthly DTR, mean temperature (TM), and soil moisture (SM) as independent variables in a Random Forest regression model to predict monthly VPD from 1980 to 2023. The model was trained using the full dataset to obtain fitted VPD values. The model was trained using the full dataset to obtain fitted VPD values VPD_{fitted} . We then conducted three sensitivity experiments—one for each independent variable. In each experiment, the tested variable was held constant at its mean value for the corresponding month during the initial period (1980–1982), while the other two variables were allowed to vary as in the original input. The difference between VPD_{fitted} and the predicted VPD from each sensitivity experiment was interpreted as the contribution of DTR, TM, or SM to the VPD change, denoted as VPD_{DTR} , VPD_{TM} , and VPD_{SM} , respectively. The detailed methodology has been added in the revised manuscript (see lines 532–553).

To enhance clarity and reproducibility, the Methods section should provide a structured and detailed explanation of each analytical step, ensuring that all procedures can be independently replicated

Res: Thank you for the suggestion. In the revised manuscript, we have restructured the Methods section and provided a detailed explanation of each analytical step to enhance clarity and ensure the reproducibility of our analysis.

Response to reviewer 2

This study examines the increase in atmospheric vapour pressure deficit since the 1980s and attempts to establish the contribution to this increase from diurnally asymmetric trends in daily maximum and daily minimum temperatures. The study uses sub-daily data from weather stations and ERA5 reanalysis, applying regression methods to characterise the relationships between atmospheric dryness (vapour pressure deficit), near surface temperature and moisture. The principal finding is that the faster rise in daily maximum temperatures compared to daily minimum temperatures over land has intensified atmospheric dryness from 1980 to 2023.

Overall, this manuscript addresses an important topic and the findings are noteworthy. However, there are areas where the manuscript could be strengthened by major revisions. My comments and suggestions are listed below.

Res: Thank you for your valuable comments and suggestions. We sincerely appreciate your thoughtful feedback, which has helped improve the quality and clarity of our manuscript. Please find our point-by-point responses below addressing the comments you suggested.

1. The results prompt important questions that merit further exploration, such as “Did VPD decline during 1950-1980 and was it driven by diurnally asymmetric temperature trends (which had the opposite sign to the post 1980 period)?” If the analysis is not extended, then I suggest including discussion of trends and the potential applicability of these results to the pre 1980 trends. The results of the paper would be more compelling if the period prior to 1980 is included in the analysis.

Res: Thank you for this insightful comment. Following your suggestion, we extended our analysis back to the 1950s. We found that, over land, VPD significantly declined from the 1950s to the mid-1970s, reaching a minimum around 1976. This decline was primarily associated with a significant decrease in daily maximum temperatures (Tmax),

while daily minimum temperature (T_{\min}) exhibited little change. In contrast, during the period from 1977 to 2023, VPD increased substantially, associated with a faster warming of T_{\max} relative to T_{\min} . These findings reinforce the robustness of our conclusions and highlight an asymmetric effect of T_{\max} and T_{\min} on atmospheric dryness, with T_{\max} playing a more dominant role than T_{\min} in driving the change of VPD. Please refer to lines 313-327 and Supplementary Fig. 21 for details.

2. The comparison of ERA5 reanalysis and HADISD stations. I think it important to compare results from these two sources on a consistent basis. I suggest a secondary result is calculated using a subset of ERA5 grid cells which are co-located with the stations. This would enable discussion of the differences. Further, inferences could be made as to the causes of the differences. For instance, when I have done this in the past, I have found that differences in altitude between the stations and the paired ERA5 grid cells can explain a reasonable proportion of the difference in temperatures. The distribution of the stations is strongly skewed towards the USA and Europe, comment could be made about differences between stations and ERA5 in different regions. These changes would enhance confidence in the ERA5-based results and provide a robust validation framework.

Res: Thank you for this insightful suggestion. We conducted additional analysis comparing ERA5 grid cells co-located with station sites using your recommended approach. Our results confirm your findings: ERA5-Land and station-based observations showed higher correlations in North America and Europe but showed larger discrepancies in East Asia. Please see lines 69–75, Supplementary Discussion 1 and Supplementary Fig. 1 in the revised manuscript for further details.

Our study utilizes both sub-daily data from stations and ERA5 reanalysis, not primarily for direct comparison between the two datasets at the sub-daily scale, but rather to provide complementary insights. For example, in the “Interannual Analysis” section, we present results from both stations and ERA5 reanalysis, and both consistently

support our key finding that T_{\max} and T_{\min} have different impacts on daily-mean VPD. Additionally, we have included a brief explanation early in the main text to clarify the rationale for using both datasets (lines 69–75 in the revised manuscript). These revisions are intended to guide readers toward understanding the complementary roles of the two datasets in reinforcing the study’s central conclusions, rather than focusing on a direct comparison between them.

It is noteworthy that the sub-daily performance of ERA5 reanalysis has been previously evaluated by one of our co-authors (Dai, 2023; 2024), showing its reliability in capturing diurnal temperature and humidity variations in most regions. Therefore, while not the focus of our current study, we considered ERA5-Land data appropriate for supporting large-scale VPD assessments in combination with station records.

References

- Dai, A. (2023), The diurnal cycle from observations and ERA5 in surface pressure, temperature, humidity, and winds, *Climate Dynamics*, 61(5), 2965-2990.
- Dai, A. (2024), The diurnal cycle from observations and ERA5 in precipitation, clouds, boundary layer height, buoyancy, and surface fluxes, *Climate Dynamics*, 62(7), 5879-5908.

3. Regression models. I would like to see more results on the quality of the fit of the models (mainly in SI) and I think it essential that the limitations of the models are discussed in the Conclusion section. Linear regression models are used while the relationship between temperature and SVP is strongly non-linear. How does this affect the results?

Res: In the revised manuscript, we have improved the ridge regression approach by implementing several robustness checks. Specifically, we use the Durbin–Watson statistic to assess autocorrelation, determine the optimal regularization parameter (λ) via leave-one-out cross-validation, and apply a nonparametric bootstrap method to quantify the statistical significance of the regression coefficients. These improvements allow for more reliable inference by avoiding assumptions of normally distributed

residuals and better accounting for sampling variability.

We agree that the relationship between temperature and SVP is strongly non-linear. In the revised manuscript, we have addressed this concern by applying a Random Forest model combined with the SHAP (SHapley Additive exPlanations) framework to quantify the relative importance of environmental variables on interannual VPD variations. This machine learning framework is capable of capturing non-linear relationships, mitigating multicollinearity, and accounting for interaction effects, thereby enhancing the robustness of our results. Please see Figs. 1 and 2 in the revised manuscript.

4. The Conclusion section could be improved with a critical assessment of the study findings versus recent literature and a more in-depth discussion of the relevance of these results for climate impacts such as drought and wildfires. This would underscore the relevance of the findings and their broader implications and, in my opinion, is particularly relevant for a journal such as Nature Communications.

Res: Thank you for your insightful suggestion. In the revised manuscript, we have expanded the discussion section to include a more critical assessment of our findings in the context of recent literature. Specifically, we discuss the potential links between the observed increase in DTR/VPD and regional droughts and wildfires drawing on previous studies. Moreover, we highlight how the rise in VPD relates to recent findings on the intensification of global drought severity driven by increased atmospheric evaporative demand (AED) (Gebrechorkos et al., 2025), for which VPD is a key component. Please see lines 382–410 in the revised manuscript. Please see a new section “Implications for drought and wildfire risk” in lines 328-355.

References

Gebrechorkos, S.H., Sheffield, J., Vicente-Serrano, S.M., Funk, C., Miralles, D.G., Peng, J., Dyer, E., Talib, J., Beck, H.E., Singer, M.B., & Dadson, S.J. (2025). Warming accelerates global drought severity. *Nature*, 642, 628-635

5. The impact of the study would be improved if results are presented in a way which quantitatively connects regional atmospheric drying (and its drivers) with regional climate impacts such as drought and wildfires. For instance, trends and regression results could be summarised for key regions of drought and escalated wildfire occurrence during 1980-2020s.

Res: In the revised manuscript, we have added new analyses to quantitatively assess the relationship between VPD and changes in DTR with drought and wildfire risk, using a standardized drought index (PDSI) and fire weather index (FWI). Our results show that PDSI over 47.7% of global land area are significantly negatively correlated with VPD during 1980–2023. Among these regions, 68.6% experienced increasing DTR and 69.3% experienced intensifying drought. Furthermore, we found that a significant positive correlation exists between FWI and VPD (93.8% of land area) as well as between FWI and DTR (85.7% of land area) during 1980–2023, highlighting the widespread influence of atmospheric drying on fire risk (lines 328-355).

Detailed comments

6. Lines 18-19. I assume the VPD, SVP, and AVP are defined/measured at the near surface (~2m). It would make the text clearer to explicitly state the level at which these terms are defined.

Res: Yes, these terms are defined and measured at the near-surface level (~2 m). We have revised the text to explicitly state this for improved clarity.

7. Lines 28-29 plus all figures and results in the paper which present results based on ERA5 reanalysis and station observations. This statement, and the presentation of results more generally in the paper, compares results from station observations and ERA5 for different regions. Therefore, it does not serve to validate the use of the ERA5 reanalysis. To validate the results based on ERA5 against observations, I suggest pairing each station with its nearest neighbour grid cell from ERA5 and

presenting the results for this subset of grid cells.

Res: Following your suggestion, we have compared the station observations with the corresponding nearest-neighbor ERA5 grid cells. The results of this comparison are now included in the Discussion section, please see Supplementary Discussion 1 and Supplementary Fig. 1 for more details. We also included a brief explanation to clarify the rationale for using both datasets, please see lines 69–75 in the revised manuscript.

8. Line 30 and other places which refer to drivers of the changes. The study would be strengthened by unpacking “global warming”. For instance, it is important to mention changes in aerosols. Trends in aerosol emissions have been key to the observed changes in diurnal temperature range (Wild 2009, Zhou et al., 2010).

Res: We have revised the relevant text to explicitly highlight the role of aerosol emissions in driving changes in DTR. Please see lines 57–59 in the revised manuscript.

9. Lines 58-61. Results from many regression models are presented in the main paper and in supplementary information. It might help the reader to outline the different methods and results in this paragraph. This would help by outlining the structure of the paper and provide a high-level guide to the how the research question is addressed.

Res: We have added a summary of the paper's structure, including the main objectives and methods used in each section, to provide a clearer roadmap for the reader. Please see the last paragraph of the Introduction in the revised manuscript.

10. Line 65 and other places where formulae are disclosed. It would help the reader to define the equations and reference them with numbers throughout. This will avoid repeating the same formula through the manuscript.

Res: We have defined the key equations and assigned them numbers, which are consistently used throughout the manuscript to avoid repetition and enhance clarity.

11. Line 69. I don't understand the phrase "large-scale characteristics of the HadISD data". This dataset is a collection of point measurements. Are you saying that the discontinuities in hydrometer measures are localised and small in number? It would help clarity to be more specific in the text.

Res: Yes. What you understand is right, that station sampling uncertainties are essentially negligible for large-scale characteristics of RH trend. We have revised related expressions. Please see lines 80–83

12. Lines 70-71. Are you referring to HadISD data or ERA5 reanalysis? I suggest you review the manuscript text to ensure it is clear throughout which results are referred to.

Res: This statement refers to the ERA5-Land dataset. We have reviewed the manuscript to ensure that all references to datasets and corresponding results are clear.

13. Lines 78-79 and use of regression methods generally. It could strengthen the robustness of the study to explore the assumptions of the regression models in greater detail. For instance, you could comment on autocorrelation and the distribution of residuals.

Res: In the revised manuscript, we have improved our methodology to enhance the robustness of the results. Specifically, we used the Durbin–Watson statistic to assess the presence of autocorrelation in the residuals. Most values were close to 2, indicating a limited impact of autocorrelation (see Supplementary Figs. 4b, 5b, 9b and 11b). Furthermore, we refined our approach by determining the optimal regularization parameter (λ) using leave-one-out cross-validation (LOOCV), which provides a data-driven and unbiased method for model parameter selection. To quantify the uncertainty and statistical significance of the ridge regression coefficients, we employed a nonparametric bootstrap method. This approach does not assume normally distributed residuals, thereby strengthening the robustness and reliability of our inferences.

Relevant details have been added to the Methods sections (lines 452–489).

14. Line 88. *It would help clarity in the main text to explain the measure you use to quantify contribution.*

Res: In the revised manuscript, we have clarified in the main text (lines 135–137) that the contribution is quantified based on the relative magnitude of the absolute SHAP values ($|\text{SHAP value}|$), which represent the importance of each predictor in the Random Forest model.

15. Figure 1 (and other figures too). *It would help clarity to have descriptive panel titles within the figures.*

Res: We have added descriptive panel titles figures in the revised manuscript to enhance clarity and readability.

16. Line 97. *It took me in a few moments to understand the following: “Spatial distribution of ridge regression coefficients (R.c., see Methods) of VPD with respect to DTR, with DTR, soil moisture (SM), and mean air temperature as independent variables.” I think it would be clearer to define an equation for each regression in the text and cross-reference the equations here and elsewhere.*

Res: Thank you for the helpful suggestion. We have now defined an equation for each regression model in the main text and provide corresponding equation numbers. These equations are cross-referenced here and throughout the manuscript to improve clarity.

17. Line 106. *I cannot see where the non-significant grid cells and stations are located. Have they been hatched or masked in a specific colour? Further, it would be informative to quote the proportion of stations and grid cells which are significant in each panel. I have the same comment for Figure 2 (lines 148-149).*

Res: In the revised manuscript, the station or grid cells without any significant ridge regression coefficients have been masked in light grey. The proportion significant stations/ land areas percent is now given in the figure caption of Figs. 1 and 2. Bootstrapping was used to evaluate the significance of each ridge regression coefficient.

18. Figure 2. It could help interpretation to change the colour scale. Differences between regional results would be clearer. Results located in the 3 vertices of the triangle are clear enough, but colours towards the centre of the triangle are not so clear. I suggest you try using colours for discrete categories (e.g., no significant contribution from Tmx, Tmn, sm; Tmx only; Tmn only; etc) rather than a continuous colour scale.

Res: Thank you for your suggestion. We agree that some of the colors in the original figure were not sufficiently clear for interpretation. In the revised manuscript, we addressed this by replacing the color scale with a control factor-based classification, where the factor with the highest absolute SHAP value is shown. We hope this improves clarity. The figure using colors for discrete categories, as you suggested, has been included in the Supplementary Information for reference (see Supplementary Figs. 4, 5, 9 and 11).

19. Line 215. I assume that the validation set is the remaining 20% - I suggest you explicitly say this to help clarity.

Res: In the revised manuscript, we have improved the method by using all available data to train the Random Forest model. The model performance is evaluated using the out-of-bag (OOB) R^2 and OOB mean squared error (MSE), which provides an unbiased estimate of the predictive accuracy without the need for a separate validation set.

20. Line 272-274. This sentence requires a more nuanced discussion. Mention could be made of: trends prior to 1980, actual projections from CMIP6, the drivers of diurnal asymmetry in warming (particularly regional changes in aerosols and solar dimming/brightening).

Res: In the revised manuscript, we have expanded this sentence and included a more nuanced discussion. Please see lines 382-409 in the revised manuscript.

21. Line 280. State explicitly the time frequency of the data used.

Res: The time frequency of the HadISD data ranges from 6-hourly to hourly. We have

clarified this point explicitly in the revised manuscript.

22. Line 290. Could you state explicitly the depth of the layer used?

Res: It is measured at a depth of 30 cm from the topsoil layer. We have explicitly stated this in the revised manuscript.

23. Line 300. Why did you use soil moisture between 0-1m depth? It appears inconsistent with the topmost layer used from Fluxnet (which I assume is shallower than 1m depth).

Res: Thank you for pointing this out. To ensure consistency with the soil moisture layer used from Fluxnet, we used the top two layers of soil moisture from ERA5-Land, corresponding to a depth of 0–28 cm, which closely matches the Fluxnet SWC measurements. Please see lines 430-431.

24. Line 327: Please justify use of the threshold of 3.

Res: In the revised manuscript, we have improved the methodology. Specifically, the optimal regularization parameter (λ) is now determined using leave-one-out cross-validation (LOOCV), rather than the original method. This ensures a more robust selection of λ .

25. Line 370. What metrics did you use to validate the random forest? I would expect to see several metrics such as RMSE, MSE, area under curve (AUC), out-of-bag (OOB). Key results from this in SI would demonstrate the robustness of the study.

Res: In the revised manuscript, we used OOB R^2 and OOB MSE to evaluate the performance of the random forest model. The spatial distributions of these metrics are provided in the Supplementary Information, please see Supplementary Fig. 16.

26. Paragraph at line 372. What quantitative techniques were used to interpret and explain the random forest models? For instance, did you use SHAP values?

Res: Thank you for your suggestion. While we used the RF-SHAP framework in the interannual analysis to interpret the random forest models, the long-term effect of DTR on VPD was analyzed using a control variable approach. Specifically, we used monthly

DTR, air mean temperature (TM), and soil moisture (SM) as independent variables in a Random Forest regression model to predict monthly VPD from 1980 to 2023. The model was trained using the full dataset to obtain fitted VPD values (VPD_{fitted}). We then conducted three sensitivity experiments—one for each independent variable. In each experiment, the tested variable was held constant at its mean value for the corresponding month during the initial period (1980–1982), while the other two variables were allowed to vary as in the original input. The difference between VPD_{fitted} and the predicted VPD from each sensitivity experiment was interpreted as the contribution of DTR, TM, or SM to the long-term change in VPD. The detailed methodology has been added to the revised manuscript (see lines 532–553).

27. References – You could consider citing Zhou et al. (2010)
<https://link.springer.com/article/10.1007/s00382-009-0644-2>

Res: This reference is indeed relevant and helpful. We have cited it in both the Introduction and Discussion sections of the revised manuscript.

28. SI Figure 1 caption. I found the statement “over land areas during” confusing because the station observations are point measurements and are not representative of large-scale land areas

Res: To be more precise, we have revised the phrase “over land areas during” to “across all stations or over land” in the revised caption of Supplementary Figure 2.

29. SI Figure 2. I would expect to see statistical significant tests applied to the gridded maps. This comment applies to all maps in the paper.

Res: We have added statistical significance information to the gridded maps throughout the revised manuscript and Supplementary Information.

30. SI Figure 2. This figure sets up the research question very nicely. You might consider moving it to be the first figure in the main paper. Panel c) shows that trends in DTR vary between regions. This is relevant to the study and something that could be commented on in the main text.

Res: Good suggestion. The spatial distribution of the trend in DTR has been moved to

Panel (a) of Fig. 1 in the revised manuscript.

31. SI Figure 2. Should 1980-2022 be 1980-2023?

Res: Yes, it should indeed be 1980–2023. We have revised it.

32. SI Figure 3. “Only the stations or grids with the regression result that passed the test of significance”. I can’t see where the excluded grid cells are. How are they shown (e.g., masked out or hatched?). It would be informative to state the proportion of stations & grid cells that are significant.

Res: In the revised manuscript, we have clarified in the figure caption that the excluded stations and grid cells—defined as those where none of the ridge regression coefficients are statistically significant—are masked in light gray. The proportion of such stations and grid cells is also stated in the caption.

Response to reviewer 1

Thank you for addressing my previous comments. The current version shows substantial improvement. Please consider the following minor comments below:

Res: Thank you for your valuable comments. We sincerely appreciate your feedback which has helped improve the quality and clarity of our manuscript. Please find our point-by-point responses below.

Line 63-65: To improve clarity and precision, consider revising lines 63-65 to: "Given the potential complex interactions and non-linear relationships among the examined climatic factors, we utilized Random Forest modeling as a complementary approach." Additionally, include one or two sentences such as: "This approach delivers robust predictive performance by effectively capturing intricate, non-linear data patterns that traditional linear models may not adequately address. Furthermore, Random Forest modeling provides feature importance rankings, enabling a clearer understanding of each climatic factor's relative contribution." Please note that I am striving to align the writing more closely with addressing the research challenges, so please adjust as appropriate based on the specifics of the research work.

Res: We appreciate your helpful suggestion. We have revised the sentence as recommended to improve clarity and precision. The updated expression can be found in lines 59-64 of the revised manuscript.

Line 76-77: Please add a reference for the calculation of vapor pressure deficit (VPD). The method described in Allen et al. (1998) is widely cited and would be appropriate to include.

Allen, R. G., Pereira, L. S., Raes, D., & Smith, M., 1998. Crop evapotranspiration Guidelines for computing crop water requirements, FAO Irrigation and Drainage Paper 56, Food and Agriculture Organization of the United Nations.

Res: Done. The reference to Allen et al. (1998) has been added.

Line 78-96: I suggest separating this section into two paragraphs: one focusing on the explanation of VPD trends and the other on DTR trends.

Res: Good suggestion. We have separated this section into two paragraphs as recommended.

Line 118-122: Eqn (2) and Eqn (3): The current formulation of these equations is not rigorous. As written, they imply that the authors aim to predict VPD, whereas the actual purpose appears to be assessing the relative importance of the explanatory variables. The authors might consider using a notation such as “ $VPD \sim f(DTR, TM, SM)$ ” and “ $VPD \sim f(T_{max}, T_{min}, SM)$ ” to clarify that VPD is simply the response variable rather than a forecast target.

Res: We agree with your comment and have revised all the original equal sign to “ \sim ” in Equations (2) – (6).

Given that the relationships among variables are already explained in Lines 118–120, and the number of variables is relatively small, the authors could also consider omitting the equation expressions.

Res: We agree that the number of variables is relatively small. However, results from many regression models are presented in the main paper, and defining the equations with numbers allows us to reference them throughout the text and caption without repeatedly writing out the full formulas. This approach improves clarity and conciseness. Please note that this expression was strongly suggested by another reviewer in the previous review round, and we have found it to be both clear and helpful. Therefore, we have retained our current expression in this section.

In addition, please use ‘ T_{mean} ’ instead of ‘ TM ’ throughout the manuscript to ensure consistency with the notation for T_{min} and T_{max} .

Res: All instances of “TM” have been replaced with “Tmean” in the revised manuscript and figures.

Line 142-146: Similarly, if the research objective can be clearly articulated in the text, it may not be necessary to present the equations here, as they are more appropriate for the Methods section.

Res: These equations are also referenced multiple times in the main text and figure captions. To improve clarity and conciseness, we have retained our current expression.

Line 204: ‘Fig. 2b-2c?’

Res: It should be Fig. 2b, 2c, we have revised it.

Line 490: Please clarify the proportion of data used for training and testing. Additionally, did the authors perform cross-validation? If so, please specify the number of folds used.

Res: Thank you for your comment. In our study, we trained the random forest model using the entire dataset without an explicit train–test split. Instead, we employed the out-of-bag (OOB) validation method, which is an internal cross-validation procedure inherent to random forest. For each tree in the forest, approximately 63% of the samples were randomly selected (with replacement) for training, while the remaining ~37% (OOB samples) were used for testing. This process is repeated across all trees, and the OOB predictions are aggregated to calculate performance metrics such as OOB R^2 and OOB MSE.

We did not perform k-fold cross-validation, as the OOB approach provides an unbiased estimate of the generalization error for random forest models.

Fig. 1: Please note that some text elements in the figure are overlapping. Make sure the figure is clear and fully legible.

Res: Thank you for pointing this out. We have revised the text in Figure 1 to avoid overlapping and ensure full legibility.

Response to reviewer 2

I am happy that my comments have been addressed and find the manuscript substantially improved. I have no further comments to make on the manuscript.

Res: Thank you very much for your kind support and the constructive feedback. Your comments have been extremely helpful in improving the quality of our manuscript. We sincerely appreciate your time and efforts throughout the review process.